# The Long-Term Pannexin 1 Ablation Produces Structural and Functional Modifications in Hippocampal Neurons

**DOI:** 10.3390/cells11223646

**Published:** 2022-11-17

**Authors:** Carolina Flores-Muñoz, Francisca García-Rojas, Miguel A. Pérez, Odra Santander, Elena Mery, Stefany Ordenes, Javiera Illanes-González, Daniela López-Espíndola, Arlek M. González-Jamett, Marco Fuenzalida, Agustín D. Martínez, Álvaro O. Ardiles

**Affiliations:** 1Centro Interdisciplinario de Neurociencia de Valparaíso, Facultad de Ciencias, Universidad de Valparaíso, Valparaíso 2360102, Chile; 2Centro de Neurología Traslacional, Facultad de Medicina, Universidad de Valparaíso, Valparaíso 2341386, Chile; 3Programa de Doctorado en Ciencias, Mención Neurociencia, Universidad de Valparaíso, Valparaíso 2340000, Chile; 4Centro de Neurobiología y Fisiopatología integrativa, CENFI, Instituto de Fisiología, Universidad de Valparaíso, Valparaíso 2340000, Chile; 5Escuela de Ciencias de la Salud, Universidad de Viña del Mar, Viña del Mar 2572007, Chile; 6Escuela de Tecnología Médica, Facultad de Medicina, Universidad de Valparaíso, Valparaíso 2529002, Chile; 7Centro de Investigaciones Biomédicas, Escuela de Medicina, Facultad de Medicina, Universidad de Valparaíso, Viña del Mar 2529002, Chile; 8Escuela de Química y Farmacia, Facultad de Farmacia, Universidad de Valparaíso, Valparaíso 2360102, Chile; 9Centro Interdisciplinario de estudios en salud, Escuela de Medicina, Facultad de Medicina, Universidad de Valparaíso, Viña del Mar 2572007, Chile

**Keywords:** pannexin 1, actin cytoskeleton, neuronal morphology, dendritic spines

## Abstract

Enhanced activity and overexpression of Pannexin 1 (Panx1) channels contribute to neuronal pathologies such as epilepsy and Alzheimer’s disease (AD). The Panx1 channel ablation alters the hippocampus’s glutamatergic neurotransmission, synaptic plasticity, and memory flexibility. Nevertheless, Panx1-knockout (Panx1-KO) mice still retain the ability to learn, suggesting that compensatory mechanisms stabilize their neuronal activity. Here, we show that the absence of Panx1 in the adult brain promotes a series of structural and functional modifications in the Panx1-KO hippocampal synapses, preserving spontaneous activity. Compared to the wild-type (WT) condition, the adult hippocampal neurons of Panx1-KO mice exhibit enhanced excitability, a more complex dendritic branching, enhanced spine maturation, and an increased proportion of multiple synaptic contacts. These modifications seem to rely on the actin–cytoskeleton dynamics as an increase in the actin polymerization and an imbalance between the Rac1 and the RhoA GTPase activities were observed in Panx1-KO brain tissues. Our findings highlight a novel interaction between Panx1 channels, actin, and Rho GTPases, which appear to be relevant for synapse stability.

## 1. Introduction

Panx1, a member of the gap-junction protein family, mediates the movement of ions and small molecules between intracellular and extracellular compartments, contributing to paracrine communication in mammalian cells [1,2,3,4,5]. Panx1 channels exhibit two modes of activity with different properties of ion and solute permeability [6]. At negative potentials, Panx1 channels show a constitutive small pore ion channel activity, characterized by low conductance, slight anion selectivity (chloride ions), and outwardly rectifying currents at depolarizing potentials [7,8,9,10]. These channels also exhibit a large pore conformation with high conductance mediating a non-selective ionic flux responsible for the permeation of ATP and other metabolites [11,12]. The release of ATP through Panx1 channels is triggered by intense or chronic neuronal activity [13,14], high concentrations of external potassium [15], mechanical stress [11], low oxygen conditions [16,17], ionotropic and metabotropic receptor signaling (NMDAR [13,18], P2X7R [19], α1AR [20], α7-nAChR [21]), and the caspase-dependent cleavage of its carboxy-terminal [12,22]. However, the precise contribution of Panx1 channels to the neuronal function under resting activity remains unclear.

Aberrant Panx1 activity has been implicated in several conditions affecting the central nervous system (CNS) [4,23], including ischemia [24,25], epilepsy [26,27,28], and Alzheimer’s disease [29]. Nevertheless, Panx1 channels also mediate physiological processes in the brain. For example, in the adult mouse brain, Panx1 absence or blockade increases the excitatory synaptic transmission and modifies the threshold for the induction of synaptic plasticity [30,31]. Furthermore, during early postnatal development, Panx1 ablation promotes neurite outgrowth, dendritic spine development, and network assembly [32,33]. Thus, the fine regulation of the Panx1 expression and activity could be critical for the proper functioning of neurons and the formation of brain circuits. Therefore, it is intriguing to understand the impact of a long-term Panx1 ablation on these processes and on the underlying structural plasticity.

It is known that the polarized and highly branched morphology of neurons is crucial in establishing synaptic contacts and hence neural circuits. In fact, plastic changes in neuronal circuits are believed to be the basis of high-order brain functions such as cognition [34]. Congruently, perturbations in their regulation are associated with several synaptopathies [35]. On the other hand, most of the excitatory synapses are localized on dendritic spines, highly dynamic structures exhibiting the capacity to change their size, morphology, and density in response to neural activity [36]. Thus, the synapses are constantly subjected to activity-dependent plasticity to store information. Consequently, they are simultaneously compensated to avoid instability in the circuits preserving their plasticity and cognitive abilities. In this regard, homeostatic regulatory changes occur at the cell and circuit levels to adapt neural properties and their molecular components, providing a balanced control of the synaptic strength [37]. Several mechanisms have been reported, such as metaplasticity, synaptic scaling, and intrinsic plasticity (for review, see [38,39,40,41]), which can be executed through a variety of cellular and molecular processes depending on the development state and neuronal activity [42]. Previously, it was reported that the lack of Panx1 activity in the adult hippocampus produces metaplastic changes modifying the threshold for induction of long-term potentiation (LTP) and depression (LTD) of excitatory synapses [30]. However, this change was revealed under activity demand, and it is unknown whether Panx1 ablation elicits additional modifications to preserve neuronal activity in resting conditions. Interestingly, a recent study reported that early postnatal Panx1 ablation prevents the homeostatic adjustment of presynaptic strength upon chronic inactivity [43].

Here, we show that the constitutive global Panx1 deletion in the adult mouse promotes several changes in excitability and synaptic transmission and extensive structural modifications in CA1 hippocampal neurons. Panx1-KO CA1 neurons exhibited enhanced excitability, higher dendritic arborization, enhanced spine maturation, and increased size of the readily releasable pool (RRP) compared to the WT condition. Furthermore, these hippocampal synapses exhibited an increased size of the postsynaptic density (PSD) and an increased number of synaptic contacts at the ultrastructural level. However, the estimation of a multiplicity index revealed that the release sites or functional contacts were lower in the Panx1-KO neurons. Remarkably, such modifications correlated with perturbations in the actin cytoskeleton dynamics, as the enhanced expression of the actin-related proteins, higher expression of the activated form of the Rho GTPase Rac1, and augmented F-actin content in the hippocampal tissue of Panx1-KO mice. Altogether, our data strongly suggest that Panx1 plays a “stabilizing” role in neuronal function and morphology by modulating actin dynamics through a mechanism that involves the Rho GTPase family.

## 2. Materials and Methods

### 2.1. Animals

All animal experiments were approved by the Ethical and Animal Care Committee of the Universidad de Valparaiso (BEA064–2015 and BEA139–2019). The experiments were carried out in adult male and female homozygous from 6–9 months old (m.o.): C57BL/6 wild-type (WT) or Panx1-knockout (Panx1-KO) mice. The generation of Panx1-KO mice has been described previously [25,44]. Mice were maintained at 21 ± 1 °C, at constant humidity (55%), and in a 12/12 h dark–light cycle; with a light phase from 08:00 to 20:00, food and water were provided ad libitum.

### 2.2. Genotyping

Genotyping for Panx1-KO mice was performed using a standard PCR on genomic DNA extracted from mouse ears. The bands were visualized using the following primers [25,44]: WT_in_ (5′-GGAAAGTCAACAGAGGTACCC-3′) and WT_ex_ (5′CTTGGCCACGGAGTATGTGTT-3′). The LacZ region was visualized in combination with the corresponding WT forward primer: Panx1_lacz_ (5′-GTCCCTCTCACCACTTTTCTTACC-3′). PCR reactions were performed using 40 cycles, and amplified products were visualized in a 3% agarose gel. The WT Panx1 band (330 bp) was visible in the WT but absent in Panx1-KO mice, and the *lac*Z region for the Panx1-KO band (630 bp).

### 2.3. Electrophysiology

Animals were deeply anesthetized with 5% isoflurane (Forane, B506; AbbVie, North Chicago, IL, USA) for 5 min, decapitated when fully sedated, and brains quickly removed and submerged in ice-cold (4 °C) dissection buffer (in mM: 124 sucrose, 2.69 KCl, 1.25 KH_2_PO_4_, 10 MgSO_4_, 26 NaHCO_3_, 10 glucose). Then, coronal hippocampal slices 300 μm thick were cut with a Vibroslice microtome (model NVSLM1; WPI Inc., Sarasota, FL, USA) and recovered in a chamber with artificial cerebrospinal fluid (ACSF, in mM: 124 NaCl, 2.69 KCl, 1.25 KH_2_PO_4_, 2 MgSO_4_, 26 NaHCO_3_, 2 CaCl_2_, and 10 glucose) bubbled with a mixture of 5% CO_2_ and 95% O_2_ for 1 h at room temperature (RT, 22–24 °C). Two recording methods were employed: whole-cell patch clamp and extracellular field potentials recordings.

For whole-cell recordings, the slices were transferred into a chamber continuously perfused with bubbled ACSF (2 mL/min) at RT. Visualization of the hippocampal CA1 pyramidal neurons was achieved with infrared differential interference contrast (DIC) with an upright Nikon Eclipse FN1 microscope (Nikon Instruments, Tokyo, Japan). Excitatory postsynaptic currents (EPSCs) were recorded from the soma of pyramidal neurons from the CA1 area of dHPC clamped at a holding potential (V_h_) −65 mV using an EPC-7 amplifier (HEKA Instruments, Westfield, MA, USA) and glass pipettes (4–8 MΩ) filled with an internal solution (in mM: 131 Cs-Gluconate, 10 HEPES, 10 EGTA, 4 MgATP, 2 Glucose, 1 CaCl_2_, 8 NaCl and 0.4 Na_3_GTP, buffered to pH 7.2–7.3 with CsOH), or (in mM: 97.5 K^+^ Gluconate, 32.5 KCl, 5 EGTA, 1 MgCl × 6 H_2_O, and 10 HEPES, adjusted to pH 7.2–7.3 with KOH) for current clamp recordings. EPSCs data were elicited at 3 s intervals, filtered at 3.0 kHz, and acquired between 6.0 and 10.0 kHz using an A/D converter (ITC-16; InstruTech, Longmont, CO, USA) and stored with the Pulse Fit software (HEKA Instruments, MA, USA). Experiments started after a 5–10 min stabilization period following the establishment of whole-cell configuration. Cells that exhibited a significant change >20% in the access resistance (7–14 MΩ) were excluded from the analysis; the capacitive currents were compensated to ~70%, and neurons were accepted if the seal resistance was >1 GΩ. EPSCs were evoked by stimulation in Schaffer collaterals fibers at ~50 µm from the recording pipette in the *Stratum pyramidale*. Averages of EPSCs were obtained by repeated stimulation at 0.3 Hz. All recordings were made in the presence of picrotoxin (PTX; 10 mM; Tocris, Ann Arbor, MI, USA) and tetrodotoxin (TTX; 0.5 mM; Cayman Chemical, Ann Arbor, MI, USA) added to the ACSF as needed. The Pulse Fit generated stimulus timing signals and transmembrane current pulses. The recording analysis was made offline with the pClamp software (Clamp-fit; Molecular Devices, Silicon Valley, CA, USA).

Short-term synaptic plasticity was evaluated by a paired-pulse ratio (PPR) and use-dependent depression. The paired-pulse stimulus was applied at four different intervals (30, 70, 100, and 300 ms) and expressed as (R2/R1), where R1 and R2 are the peak amplitudes of the first and second postsynaptic responses, respectively. Spontaneous excitatory postsynaptic currents (sEPSCs) were continuously recorded at −65 mV for 30 min under PTX. In addition, miniature excitatory postsynaptic currents (mEPSCs) were recorded after adding TTX to the bath. We used a previously described approximation to estimate the size of the readily releasable pool (RRP) of vesicles [45,46,47]. In addition, the use-dependent synaptic depression was analyzed using 10 Hz bursts of 25 stimuli every 60 s (~3 V, 200 ms). The amplitudes of EPSCs evoked by the train were measured and summed and were made of the cumulative EPSC analysis. Finally, the multiplicity index was estimated as previously described [48,49]; it was calculated as the mean amplitude of action potential (AP) driven events (a) divided by the mean quantal size (q: mean amplitude of mEPSC recorded in TTX). The (a) values were determined for each cell, subtracting the contribution of the mEPSC to the pool of events collected in the absence of TTX, using the expression for a:(1)$$a=\frac{\mathrm{fbb}-\mathrm{fqq}}{\mathrm{fb}-\mathrm{fq}}$$
where f_b_ and f_q_ denote the mean frequency values from events recorded before and after the addition of TTX to the perfusion media, respectively, and (b) is the mean amplitude of both action-potential-driven sEPSCs and mEPSCs.

For extracellular field recordings, the slices were placed in a submersion recording chamber perfused with ACSF (2 mL/min). Field excitatory postsynaptic potentials (FPs), basically AMPAR-mediated FPs (AR-FPs), were evoked by stimulating Schaffer collaterals fibers with 0.2 ms pulses delivered through a bipolar theta glass stimulating electrode (TG200–4; Warner Instruments, Hamden, CT, USA) and were recorded in CA1 *Stratum radiatum* using glass electrodes filled with ACSF (1 MΩ). FPs were amplified, low-pass filtered (1700 Differential AC Amplifier; A-M Systems, Sequim, WA, USA), and then digitized (NI PCI-6221; National Instruments, Austin, TX, USA) for measurement. Baseline responses were recorded at 0.033 Hz. Next, the NMDAR EPSPs (NR-FP) were isolated by applying CNQX (10 mM; Tocris, PA, USA) in ACSF containing 2 mM Ca^2+^ and 0.1 mM Mg^2+^. After 30 min of CNQX pre-incubation, NR-FPs were recorded. Finally, basal synaptic transmission was assayed by determining the input–output relationships from FPs generated by gradually increasing the stimulus intensity; the input was the peak amplitude of the fiber volley (FV), and the output was the initial slope of the FP. All data acquisition and analysis were performed with custom-made written Igor Pro 6.3A (Wavemetrics Inc., Lake Oswego, OR, USA). Representative traces are an average of four consecutive responses.

### 2.4. Golgi Staining

Neuron morphology and dendritic spines were measured using the FD Rapid GolgiStain Kit according to the manufacturer’s guidelines (FD NeuroTechnologies, Columbia, MD, USA). Briefly, the dissected mouse brains were immersed in Solution A/B for two weeks in dark conditions at RT. The brains were placed in Solution C for 24 h in the dark. Afterward, coronal slices 150 μm thick were obtained using a semi-automatic cryostat microtome (Kedee KD-2950, ZJ, CN) at −20 °C and mounted on gelatin-coated microscope slides with Solution C. The sections were allowed to dry naturally at RT, then placed in a mixture of Solution D/E for 10 min. Next, sections were rinsed twice in Milli-Q water for 4 min each time. Finally, the sections were dehydrated, cleared in xylene, and mounted using Permount mounting media (Sigma-Aldrich, Burlington, MA, USA).

For morphometric analysis, digital images were taken of individual well-impregnated pyramidal neurons of the CA1 region using a Leica DM500 microscope (Leica Microsystems GmbH, Westlar, Germany) equipped with a 40X objective and ICC50W digital camera (Leica Microsystems GmbH). Neurons were then drawn with the aid of a camera lucida (Leica tracing device L3/20; Leica Microsystems GmbH) attached to the microscope (Leica Microsystems GmbH) and digitalized in 1.200 × 1.200 dpi resolution to morphometric Sholl analysis using the Neuroanatomy and Simple neurite tracer plugins of the Fiji software (version 1.53q, [50]). In brief, a series of concentric spheres were drawn with intersections at 20 μm interval distance points starting from the cell body to calculate the number of dendrites’ intersections. This analysis was performed separately for basal and apical dendrites. Four parameters were used to determine dendritic morphology and complexity: (1) the number of dendrites; (2) the total dendritic length, including all dendritic branches; (3) the number of dendritic branches; and (4) the branching order. All the morphological analyses were performed blind to the experimental conditions.

For spines’ analysis, dendritic segments from primary and secondary branches of 10 to 20 µm in length were selected randomly. Images were acquired using a Leica DM500 microscope (Leica Microsystems GmbH) equipped with a 63X oil HCPL APO objective (N.A 1.40) and ICC50w digital camera (Leica Microsystems GmbH) with a 1388  × 1036 pixels resolution and identical acquisition settings between compared samples.

Spine density was defined as the number of spines per 10 µm. Dendritic spines were classified according to the following parameters described in Golgi-stained neurons [51]: Long, thin without-head protrusions of more than 2 µm long were classified as filopodia; wide-head (>0.6 µm width) and short protrusions (<1 µm) were classified as mushroom-spines; wide-head/without neck protrusions (length/width ratio < 1 µm) were classified as stubby-spines; thin- and short (<2 µm)-headed protrusions were classified as thin-spines. Cup-shaped protrusions were classified as branched spines. Additionally, filopodia, stubby, and thin morphologies were classified as immature spines while mushroom and cup-shaped morphologies were classified as mature spines. The spine parameters were analyzed using Image J (version 1.49v; NIH, USA).

### 2.5. Electron Microscopy

Three male mice were transcardially perfused with a mixture of 2% paraformaldehyde (PFA) and 2.5% glutaraldehyde in 0.1 M sodium cacodylate buffer pH 7 and maintained in the same mixture overnight at 4 °C. Brain tissues were sectioned at 500 µm and post-fixed in 1% osmium tetroxide in cacodylate buffer, treated with 2% aqueous uranyl acetate, and dehydrated in an acetone battery. Then, samples were pre-embedded in Epon/acetone 1/1 overnight and then in Epon for 6 h. The inclusion was made in silicone molds with pure Epon resin and polymerized at 60 °C for 48 h. Ultrathin sections 90 nm were obtained with the ultramicrotome (Leica Ultracut R; Leica Microsystems GmbH) and collected on 300 mesh copper grids (Ted Pella Inc., Redding, CA, USA). Sections were further stained with a solution of uranyl acetate and lead citrate. Grids were unselectively sampled and the images collected of all synapses encountered in the CA1 *Stratum radiatum* area about 100 mm from the pyramidal cell layer were examined in a transmission electron microscope Philips Tecnai 12 operated at 80 kV (FEI/Philips Electron Optics, EIN, NL) equipped with a digital micrograph Megaview G2 CCD camera (Olympus-SIS, MU, GE). For the structural analysis, highly magnified electron micrographs (26,500X) of the hippocampal CA1 region were analyzed using Image J software following double-blind quantification. Assymetric synapses were differentiated by the thickening of the PSD and the membrane stretching of the active zone (AZ) directly opposite to the PSD. Docked vesicles were defined as those in direct contact with the AZ. PSDs and docked vesicles were counted manually. Freehand tools were used to analyze: the total number of synaptic vesicles; the number of docked vesicles; the number of vesicles within the active zone; PSD length; and the number of synaptic contacts. The procedure was performed by investigators blinded to the genotype.

### 2.6. Subcellular Fractionation-Synaptoneurosome Isolation

Synaptoneurosomes were extracted from the hippocampus of 6–9 m.o adult mice, as we previously reported [52]. Hippocampi were homogenized using a Dounce Tissue Grinder in ice-cold (4 °C) homogenization buffer (in mM: 320 sucrose, 4 HEPES, 1 EGTA, and protease and phosphatase inhibitor’s cocktail, buffered to pH 7.4). The homogenate was centrifuged at 1800 rpm for 10 min at 4 °C (Beckman F0630 rotor; IN, USA), obtaining a supernatant (S1) which was collected, whereas the pellet (P1) was discarded. Then, S1 was centrifuged at 30,000 rpm for 30 min at 4 °C (Beckman S4180 rotor; IN, USA). The obtained pellet (P2) containing the membrane proteins was re-suspended in the homogenization buffer, layered on the top of a discontinuous sucrose density gradient (0.32/1.0/1.2M), and subjected to ultracentrifugation at 86,000 rpm (Beckman SW-60ti rotor; IN, USA) for 2 h at 4 °C. Afterward, the sediment and sucrose 0.32/1 M interface were discarded, whereas the material accumulated at the interface of 1.0 M and 1.2 M sucrose containing synaptoneurosome fraction was collected (SP1). SP1 was diluted with lysis buffer to restore the sucrose concentration back to 320 mM and remained on ice with gentle agitation for 30 min. Then, SP1 was centrifuged at 30,000 rpm for 30 min at 4 °C. The pellet obtained (PS1) was resuspended in a gradient-loading buffer, loaded on 0.32/1.0/1.2 M discontinuous gradient, and centrifuged at 86,000 rpm for 2 h at 4 °C. The sucrose 1/1.2 M interphase, synaptoneurosome fraction 2 (SP2), was recovered and delipidated in a delipidating buffer. Next, SP2 was diluted with a filling buffer to restore the sucrose concentration and then centrifuged at 30,000 rpm for 1 h at 4 °C. The sediment obtained (PS2) was washed with 50 mM HEPES–Na and centrifuged at 86,000 rpm for 20 min at 4 °C. The final sediment obtained (PS3), containing PSDs, was re-suspended in 50 mM HEPES–Na and homogenized. PS2 or PSD fractions were quantified for protein concentration using the Qubit^®^ Protein Assay Kit (Thermo Scientific, Bannockburn, IL, USA).

### 2.7. Western Blotting

Total proteins and synaptosomal fractions were processed for Western blotting as previously described [53]. Samples for total hippocampal proteins were homogenized in ice-cold (4 °C) lysis buffer (in mM: 150 NaCl, 10 Tris-Cl, 2 EDTA, 1% Triton X-100 and 0.1% SDS, buffered to pH 7.4) supplemented with a protease and phosphatase inhibitor cocktail (Thermo Scientific, IL, USA) using a glass Potter-homogenizator. Protein samples from slice homogenates were centrifuged twice at 14,000 rpm for 5 min at 4 °C. Protein concentration was determined using the Qubit^®^ Protein Assay Kit (Thermo Scientific, IL, USA).

For both cases, 40 µg of protein per lane were resolved by 10% SDS-PAGE for synaptic proteins and 12% SDS-PAGE for actin-proteins followed by immunoblotting on polyvinylidene fluoride (PVDF membranes; BioRad, Hercules, CA, USA) and probed with specific antibodies against Panx1 (rabbit anti-Panx1; ABN242 Merck; 1:1000); PSD95 (mouse anti-PSD95; MAB1596 Merck; 1:1000); SAP102 (mouse anti-SAP102; 75–068 NeuroMAB; 1:500); Synaptophysin (goat anti-SYP; sc-9116 Santa Cruz; 1:2000); Syntaxin (mouse anti-STX; ab3265 Abcam; 1:1000); Dynamin 1 (mouse anti-Dyn1; sc-12724 Santa Cruz; 1:1000); Dynamin 3 (rabbit anti-Dyn3; ab3458 Abcam;1:1000); N-WASP (mouse anti-WASP; sc-13139 Santa Cruz; 1:2000); Arp3 (rabbit anti-Arp3; 07272 Merck; 1:1000); Drebrin (mouse anti-Drebrin; sc-374269 Santa Cruz; 1:1000); Wave 1 (rabbit anti-Wave 1; ab50356 abcam; 1:1000); Rac1 (rabbit anti-Rac1; sc-217 Santa Cruz; 1:1000); RhoA (mouse anti-RhoA; sc-418 Santa Cruz; 1:1000); Cdc42 (mouse anti-Cdc42; sc-8401 Santa Cruz; 1:1000); and GAPDH (mouse anti-GAPDH; sc-47724 Santa Cruz; 1:1000).

Blots were then washed with Tris Saline buffer with 0.2% Tween-20 (TBST, buffered to pH 7.4) and incubated with horseradish peroxidase (HRP)-conjugated secondary antibodies: anti-mouse HRP (115-035-003 Jackson Immunoresearch; 1:5000), anti-rabbit HRP (111-035-003 Jackson Immunoresearch; 1:5000), or with anti-goat HRP (705–035-003 Jackson Immunoresearch; 1:5000) was performed for 1 h at RT. Later, membranes were washed with TBST twice. Finally, Western blot was developed by chemiluminescence using ECL reagent (Pierce; Thermo Scientific, IL, USA) and detected using the image acquisition system Epichemi3 Darkroom (UVP Bioimaging System, Upland, CA, USA). Immunoreactive bands were scanned and densitometrically quantified using the Image J software (version 1.49v; NIH, MD, USA). Total and synaptosomal fractions of protein data were normalized to GAPDH.

### 2.8. Quantification of F-actin Staining

Hippocampal slices previously stabilized with ACSF bubbled with a mixture of 5% CO_2_ and 95% O_2_ for 1 h at RT were fixed with 4% PFA and 15% sucrose overnight at 4 °C. After that, slices were cut into 25 µm sections using a cryostat (Leica CM1900; Leica Microsystems GmbH) and permeabilized with 0.1% Triton X-100 for 30 min. Tissue sections were incubated with 100 nM of rhodamine-phalloidin (PHDR1; Cytoskeleton Inc., Denver, CO, USA) for 1 h. Additionally, hippocampal sections were co-stained with 4,6-diamidin-2-phenylindol (DAPI, Sigma Aldrich; 1:1000) to label nuclei. Finally, samples were visualized and acquired in an upright confocal laser-scanning Nikon Eclipse C1 Plus microscope (Nikon Instruments, NY, USA) using a Nikon 4X Plan Apo (N.A 0.20) and Nikon immersion-oil 40X (N.A 1.30) objectives. DAPI was excited by 405 nm laser light, and rhodamine-phalloidine was excited by 555 nm laser light. Single confocal images were acquired with the EZ-C1 software (Nikon Instruments, NY, USA) with a 1024  × 1024 pixels resolution and identical acquisition settings between compared samples.

The fluorescence intensity was measured in the *Stratum radiatum* layer from the CA1 area using the Image J software (version 1.49v; NIH, MD, USA).

### 2.9. F-actin/G-actin Assay

The relative amounts of filamentous (F-actin) and monomeric actin (G-actin) were quantified as previously described [54]. Briefly, hippocampal slices were previously stabilized with ACSF bubbled with a mixture of 5% CO_2_ and 95% O_2_ for 1 h at RT; then were lysed and homogenized in conditions that stabilize F-actin and G-actin using an F/G actin commercial assay (G/F actin in vivo assay Biochem kit, BK037; Cytoskeleton Inc., CO, USA). The homogenates extracts were ultracentrifuged at 86,000 rpm (Beckman SW-60ti rotor; IN, USA) for 1 h at 37 °C to separate F-actin (pellet) and G-actin (supernatant) fractions. Next, the F-actin pellet was resuspended in a depolymerizing buffer (BK037; Cytoskeleton Inc., CO, USA). Finally, the F- and G-actin fractions were diluted in loading buffer (50 mM Tris–HCl, 2% SDS, 10% glycerol, 1% β-mercaptoethanol, and bromophenol blue). All samples were resolved on 12% SDS-PAGE and detected via Western blot analysis using specific antibodies against all the actin isoforms (rabbit anti-actin, BK037 Cytoskeleton Inc.; 1:500). Finally, protein bands were visualized by ECL (Pierce; Thermo Scientific, IL, USA) and detected using the image acquisition system Epichemi3 Darkroom (UVP Bioimaging System, Upland, CA, USA). The F/G-actin ratio in hippocampal slices was calculated according to the densitometric analysis using the Image J software (version 1.49v; NIH, MD, USA).

### 2.10. Rho GTPases Pull-Down Assay

The activation of Rac1/Cdc42, and RhoA, respectively, were measured using a Rho GTPase Activation Assay Combo Biochem Kit (BK030; Cytoskeleton Inc., CO, USA), according to the manufacturer’s recommendations. Briefly, hippocampal slices were stabilized with ACSF bubbled with a mixture of 5% CO_2_ and 95% O_2_ for 1 h at RT; then were lysed and incubated at 4 °C on a rotator for 1 h with Rac/Cdc42 (PAK1 PAK-binding domain) or RhoA (Rhotekin-binding domain) beads which bind specifically to the GTP-bound, and not the GDP-bound forms of the Rho GTPases. Next, the agarose beads were collected by centrifugation at 14,000 rpm for 3 min at 4 °C and washed. The immunoprecipitated samples were diluted in loading buffer (50 mM Tris–HCl, 2% SDS, 10% glycerol, 1% β-mercaptoethanol, and bromophenol blue). All samples were resolved on 12% SDS-PAGE and detected via Western blot analysis using specific antibodies against Cdc42 (mouse anti-Cdc42, ACD03 Cytoskeleton; 1:250), Rac1 (mouse anti-Rac1, ARC03 Cytoskeleton Inc; 1:500), and RhoA (mouse anti-RhoA, ARH05 Cytoskeleton Inc.; 1:500). Finally, protein bands were visualized by ECL (Pierce; Thermo Scientific, IL, USA) and detected using the image acquisition system Epichemi3 Darkroom (UVP Bioimaging System, Upland, CA, USA). Rac/Cdc42 or RhoA activation values were expressed as the ratio of Rac1-GTP, Cdc42-GTP, or RhoA-GTP against the total proteins of each Rho GTPases in the crude extract. The ratio was calculated according to the densitometric analysis performed using the Image J software (version 1.49v; NIH, MD, USA).

### 2.11. Primary Neuronal Culture and Transfection

Primary hippocampal neurons were obtained from postnatal day 0–3 (P0-P3) WT and Panx1-KO pups and prepared from a modified protocol [55]. Briefly, hippocampi were rapidly dissected from the brain in 2 mL HBSS-1X buffer (in mM: 1 sodium pyruvate, 10 HEPES, and 20% glucose, buffered to pH 7.4) at 4 °C. Neurons were acutely dissociated at 37 °C for 20 min with 0.025% trypsin (GIBCO; Thermo Scientific, IL, USA) and 2 min with 0.1% DNase (Roche, DE) solution. After that, the hippocampi were mechanically dissociated using a 1 mL pipette. Neurons were seeded on glass coverslips pre-coated with 50 μg/mL poly-D-lysine solution (Sigma-Aldrich, MA, USA) at a density of 100–200 cells per mm^2^ in plating medium (MEM with 10% FBS, 20% glucose, 1 mM sodium pyruvate, 0.5 mM Glutamax™, and 100 μg/mL penicillin/streptomycin, buffered to pH 7.2). After 1.5 h, the medium was aspirated from each well and replaced by growth media consisting of Neurobasal-A medium (GIBCO; Thermo Scientific, IL, USA) with 2% B-27 supplement (GIBCO; Thermo Scientific, IL, USA), 1 mM Glutamax™, 1 mM sodium pyruvate, and 100 μg/mL penicillin/streptomycin, and maintained at 37 °C in a humidified incubator under 5% CO_2_ atmosphere and 99% humidity. After two days, cultured neurons were treated with 2 μM cytosine-arabinoside (Ara-C; Sigma-Aldrich, MA, USA) for 24 h, to limit but not eliminate the non-neuronal cell development. Then, 50% of the growth media was slowly and gently replaced with fresh medium (same composition) and then every 3 days. In these conditions, the presence of glial cells represented nearly 30–40% of the total cell numbers.

### 2.12. Transfection and Phalloidin Staining in Cultured Neurons

At days in vitro (DIV) 13, hippocampal neurons were transiently transfected with 1 μg Panx1-EGFP and 1 μL Lipofectamine 2000 (Thermo Fisher Scientific, MA, USA) during 1 h and evaluated at DIV 14–15. The EGFP reporter gene was used to identify transfected cells. On days 14–15, the neurons were fixed and processed for F-actin staining.

To quantify F-actin, hippocampal neurons (DIV 14–15) were fixed with 4% PFA and 4% sucrose for 10 min at 37 °C and permeabilized with 0.1% Triton X-100 for 30 min. Neurons were incubated with 100 nM of rhodamine-phalloidin (PHDR1; Cytoskeleton Inc., CO, USA) for 1 h. Next, the neurons were co-stained with 4,6-diamidin-2-phenylindol (DAPI; Sigma Aldrich; 1:1000) to label the nuclei. Finally, samples were visualized and acquired in an upright confocal laser-scanning Nikon Eclipse C1 Plus microscope (Nikon Instruments, NY, USA) using a Nikon Plan Flour 40X (N.A 1.30) and Nikon SR Apo 100X (N.A 1.49) objectives. DAPI was excited by 405 nm laser light, rhodamine-phalloidine was excited by 555 nm laser light, and EGFP was excited by 488 nm laser light. Single confocal images were acquired with the EZ-C1 software (Nikon Instruments, NY, USA) with a 1024  × 1024 pixels’ resolution and identical acquisition settings between compared samples. The neuronal dendritic and spine morphology were analyzed using the Neuroanatomy plugin of Image J (version 1.49v; NIH, USA). The spines’ parameters used were described in Section 2.3 (see Section 2).

### 2.13. Statistics

All data were presented as mean ± standard error of the mean (SEM). All statistical analyses were performed using the GraphPad Prism 9 software (GraphPad Software Inc., San Diego, CA, USA). A Shapiro–Wilk test probed the normality of the raw data. Significant differences were tested by the two-tailed Student’s *t*-test or Mann–Whitney rank test for two sample comparison, ANOVA (one-way or two-way) followed by Tukey’s or Bonferroni’s post hoc test, or Kruskal–Wallis followed by a Dunn’s correction for multiple comparisons. For repeated measures, two-way ANOVA was followed by Bonferroni’s post hoc test. Differences were considered statistically significant at *p* < 0.05. For the analysis of the cumulative probabilities, a two-sample Kolmogorov–Smirnov test test was taken into analysis, and the difference was considered significant when *p* < 0.05.

## 3. Results

### 3.1. Panx1 Ablation Increases Neural Excitability without Affecting the Spontaneous Glutamate Release

We determined the impact of the Panx1 deletion on the excitatory synaptic strength upon basal conditions by using whole-cell current- and voltage-clamp recordings in CA1 hippocampal neurons from Panx1-KO mice and WT littermates (Figure 1). First, we examined the intrinsic electrical properties of the hippocampal pyramidal neurons (Table 1). Resting membrane potential (V_m_), input resistance (R_in_), and neuronal firing properties might contribute to neuron excitability. We observed that Panx1-KO neurons display a lower threshold for the AP discharge (Figure 1, B and C; WT versus KO * *p* = 0.0108; Mann–Whitney test). In addition, the somatic values of Rin were higher for Panx1-KO than for WT ones, whereas the resting membrane potential and the cell capacitance were not significantly different (Table 1). Next, we evaluated the voltage-gated ionic currents. We did not observe differences in the I-V curves at depolarizing potentials (Figure 1D,E; WT versus KO * *p* = 0.0430; Mann–Whitney test), but the Panx1-KO neurons showed a lower spike-frequency adaptation relative to WT neurons (Figure 1F; WT versus KO * *p* < 0.0142, ** *p* < 0.046 and *** *p* < 0.001; two-way ANOVA).

Next, we evaluated whether the variations in excitability in the Panx1-KO neurons affected the synaptic transmission. We analyzed the evoked basal synaptic transmission mediated by AMPAR and NMDAR by generating input–output curves measured as extracellular field excitatory postsynaptic potentials (FP) at different stimulus intensities (Figure 1G–Q). The slope and the fiber volley (FV) amplitudes of AMPAR- and NMDAR-mediated FPs (AR and NR-FP) were similar between WT and Panx1-KO conditions, suggesting no changes in basal synaptic transmission (Figure 1I,J,N,O). However, it is noteworthy that evoked AR- and NR-FP from Panx1-KO neurons exhibited an increased tendency to show population spikes at growing intensities (Figure 1, H and M), and differences in the number and amplitude of pop spikes (Figure 1K,L and *p*-Q; WT versus KO *** *p* < 0.001; Mann–Whitney test) revealing latent greater excitability of the Panx1-KO neurons.

To further evaluate these differences, we examined the effect of the Panx1 ablation on the release probability (Pr) in the CA3-CA1 synapses by estimating two principal forms of short-term synaptic plasticity: paired-pulse ratio (PPR) and use-dependent depression. The PPR was measured as the relative strength of the second of two consecutive synaptic events, which were inversely related to the Pr [56]. The use-dependent depression of the synaptic strength was measured as the response to a high-frequency stimulus train [46,57]. Analysis of the PPR revealed substantial differences between experimental groups (Figure 2A; * *p* = 0.074; two-way ANOVA test and Figure 2B; * *p* = 0.0144; Mann–Whitney test). Individual data at 300 ms of interstimulus interval (ISI) showed a significant decrease in the PPR in Panx1-KO neurons compared to WT, suggesting an increase in the P_r_ (Figure 2C; * *p* = 0.0144; Mann–Whitney test). In addition to modifications in the release probability, changes in neurotransmitter release can result from alterations in the size of the RRP of synaptic vesicles. Thus, we evaluated the use-dependent depression during a 10 Hz stimulus train, which is an estimation of the size of the RRP [58]. After the initial increase in the EPSC amplitude in WT animals, we observed a progressive depression, consistent with a gradual depletion of the RRP recruited by the train (Figure 2D). Interestingly, we observed persistent facilitation throughout the whole stimulatory train in Panx1-KO neurons, which was reflected in the normalized curve of EPSCs (Figure 2E; WT versus KO ** *p* = 0.017; two-way ANOVA test) and the cumulative current amplitude that was significantly different between experimental groups (Figure 2F; WT versus KO *** *p* = 0.002; two-way ANOVA test). Together, these data suggest that a decrease in the EPSC amplitude would be accounted for by a change in the P_r_ and/or a decrease in the RRP size, which seemed to be greater in Panx1-KO neurons.

Excitatory short-term synaptic plasticity, particularly synaptic depression, may also depend on postsynaptic mechanisms such as desensitization or saturation of AMPARs and/or NMDARs [59]. Hence, we tested for the contribution of postsynaptic receptors in the increased facilitation observed in Panx1-KO CA1 neurons by estimating an NMDAR- to AMPAR-mediated EPSCs ratio (NR/AR) and a PPR for AR and NR-mediated EPSCs, respectively (Figure 2G,H). We found that NR/AR was indistinguishable between groups, although NR- and AR-mediated currents were reduced in Panx1-KO mice compared to WT littermates (Figure 2G,H). Similarly, we observed that the PPR elicited by NR- and AR-mediated EPSCs were comparable between groups (Figure 2I,J).

We next analyzed the impact of the Panx1 ablation on the spontaneous synaptic activity by sEPSCs and mEPSCs. We observed that either the frequency or the amplitude of sEPSCs were similar between groups, indicating that the Panx1 ablation does not affect the basal glutamate release (Figure 3A–D). However, we found that the absence of Panx1 increases the amplitude of mEPSCs (Figure 3E; WT versus KO * *p* = 0.0104; Mann–Whitney test), but the frequency of mEPSCs was unaltered between groups (Figure 3E–H).

We also estimated a multiplicity index, an approximation to estimate the number of releasing sites and functional synaptic contacts (see Materials and Methods; Figure 3I,J) [48,49]. Interestingly, we found that the multiplicity index in Panx1-KO CA1 neurons was significantly lower than that observed in WT neurons (Figure 3J; WT versus KO * *p* = 0.0012; Mann–Whitney test), suggesting that Panx1-KO neurons could establish less functional contacts or possess fewer releasing sites.

Overall, these results strongly suggest that the spontaneous neurotransmission is unaltered in Panx1-KO CA1 neurons, but the evoked glutamate release seems prompt to respond to activity demand. Furthermore, the Panx1 ablation appears to enhance the neural excitability suggesting that Panx1-KO neurons could hold homeostatic-like mechanisms to maintain synaptic transmission.

### 3.2. Increased Dendritic Arborization and Dendritic Spine Maturity of Hippocampal CA1 Pyramidal Neurons of Panx1-KO Mice

As the Panx1 ablation seems to induce adjustments in the network connectivity and neuron excitability, we analyzed the impact of the Panx1 ablation on the morphology of Golgi-Cox-stained neurons at dendrite and dendritic spine levels (Figure 4).

A significantly higher dendritic branching and dendritic length were observed in CA1 hippocampal neurons from Panx1-KO mice compared to age-matched WT animals (Figure 4A–E). Panx1-KO neurons exhibited longer dendrites (Figure 4B; WT versus KO ** *p* = 0.002; Mann–Whitney test) with almost twice branch points (Figure 4C; WT versus KO ** *p* = 0.002; Mann–Whitney test) in both apical and basal dendrites (Figure 4D,E; WT versus KO ** *p* = 0.002; Mann–Whitney test), showing a great number of dendrites throughout the proximal to distal distance from the soma in the Stratum radiatum layer (Figure 4F; WT versus KO ** *p* = 0.002; Mann–Whitney test).

Furthermore, Sholl’s analysis also revealed that the Panx1-KO neurons displayed a significantly increased branch order indicative of a greater dendritic complexity in the apical and basal neuronal compartments (Figure 4F,G; WT versus KO *** *p* < 0.001; two-way ANOVA test). Concomitantly, the analysis of dendritic segments of Golgi-impregnated neurons (Figure 4H) revealed similar dendritic spine density (Figure 4I), but a significantly increased dendritic spine length (Figure 4J; WT versus KO ** *p* = 0.004; Mann–Whitney test) in basal dendrites between WT and Panx1-KO mice. Interestingly, we classified dendritic spines based on their morphology in mature (mushroom- and cup-shaped) and immature spines (filopodia, thin, and stubby) (Figure 4L,M). We observed that, in general, Panx1-KO neurons exhibited a greater proportion of mature-type spines than WT neurons (Figure 4K,L; WT versus KO * *p* = 0.02, ** *p* = 0.005, *** *p* < 0.001; two-way ANOVA test), indicating that the absence of Panx1 also induces significant structural changes in the spine morphology.

These data suggest that the lack of Panx1 promotes a significant modification in neuronal morphology.

### 3.3. Structural and Molecular Remodeling of the Synapses in Hippocampal CA1 Pyramidal Neurons from Panx1-KO Animals

Next, we examined the effect of the Panx1 ablation on the synaptic structure by electron microscopy (Figure 5). We found that CA1 hippocampal neurons from Panx1-KO mice frequently exhibited compound synapses, either multi-innervated spines or an axon terminal contacting multiple spines (Figure 5A,B; Table 1; Appendix A (Appendix A)). Furthermore, analysis of compound synapses revealed an increase in the proportion of multiple synaptic contacts in Panx1-KO mice compared to WT animals (Figure 5B; WT versus KO ** *p* = 0.002, *** *p* < 0.001; two-way ANOVA test). Moreover, we found a shorter PSD size in Panx1-KO neurons (Figure 5C; WT versus KO * *p* = 0.03; Mann–Whitney test). Remarkably, the number of small clear vesicles was significantly higher in Panx1-KO spines compared to WT (Figure 5D,E; WT versus KO * *p* = 0.0286; Mann–Whitney test); in both, the total vesicle docked to the axon terminal membrane as well as the docked vesicles at the active zone (Figure 5F,G; WT versus KO * *p* = 0.03; Mann–Whitney test).

Early studies have shown that Panx1 is expressed by neurons and glia [60] and it accumulates preferentially in the PSD of cortical and hippocampal neurons, colocalizing with postsynaptic proteins such as AMPAR and PSD-95 [61]. We verified these observations by isolating hippocampal synaptoneurosomes and subsequently separating them into PSD-enriched fractions and synaptic membranes devoid of PSD (SM) (Figure 5H; WT versus KO * *p* = 0.002; two-way ANOVA test). Interestingly, we detected the presence of Panx1 in both SM- and PSD-enriched fractions suggesting that Panx1 is expressed at the pre- and post-synaptic terminals. Although no apparent changes in the overall protein patterns were revealed by the Coomasie blue-stained gel electrophoresis (Appendix A), Western blot from the whole hippocampal tissue showed modifications in several synaptic proteins (Appendix A). In this regard, we found a significant increase in presynaptic proteins such as synaptophysin (SYP) and postsynaptic proteins including PSD-95 and SAP-102, in both whole hippocampal tissue (Appendix A) and synaptic enriched fractions obtained from KO mice (Figure 5I; WT versus KO *** *p* < 0.001; two-way ANOVA test). Overall, these results suggest that lacking Panx1 produces a major restructuring of the pre- and post-synaptic composition, consistent with the increase in synaptic connectivity.

Dissociated primary neuronal cultures represent a tool to recapitulate in vitro, many aspects of brain development that occur in vivo such as neuronal maturation, structural plasticity, and functional activity. We used dissociated primary neuronal cultures derived from a postnatal brain which contained a significant number of glial cells, maintaining their supportive role for neurons. To validate the role of Panx1 channels in the maturation of dendritic arbors and spines, we attempted to use a rescue strategy, by transfecting WT and Panx1-KO primary hippocampal neurons with Panx1-EGFP or with the EGFP-empty vector as control. Hippocampal neurons were transfected at DIV13 and analyzed at DIV 14–15, corresponding to the time when most spines undergo morphological and functional maturation [62,63,64,65,66]. As expected, Panx1-KO neurons exhibited longer dendrites with a higher number of branches in all the dendritic arbors compared to WT neurons (Figure 6A–D).

Interestingly, under Panx1-EGFP transfection, we observed a significant reduction in all dendritic morphological measurements in Panx1-KO and WT neurons (Figure 6A–D; WT versus KO Panx1-EGFP *** *p* < 0.001; two-way ANOVA test). These results suggest that the overexpression of Panx1 is sufficient to induce a reduction in the dendritic arbor in WT neurons and that it rescued the dendritic phenotype of Panx1-KO neurons.

To further explore the effects of the Panx1 ablation and their rescue on the dendritic spine morphogenesis, we analyzed the spine density and shape in cultured neurons to further explore the effects of the Panx1 ablation and their rescue on dendritic spine morphogenesis (Figure 7). We visualized the dendritic segments by rhodamine-phalloidin fluorescence, a toxin that specifically binds to F-actin [67]. Interestingly, we observed a higher intensity of phalloidin in both the dendritic shaft and spine from Panx1-KO (Figure 7A–C; WT versus KO *** *p* < 0.001; two-way ANOVA test), which was reduced in Panx1-KO mice transfected with Panx1-EGFP (Figure 7B,C; KO versus KO Panx1-EGFP *** *p* < 0.001; two-way ANOVA test).

The analysis of dendritic spines’ morphogenesis showed that the lack of Panx1 in the primary hippocampal neurons led to a remarkably higher spine length, but similar spine density compared to WT neurons (Figure 7D,E; WT versus KO *** *p* < 0.001; two-way ANOVA test). However, the spine length and the proportion of mature spines were significantly reduced in neurons from Panx1-KO mice transfected with Panx1-EGFP (Figure 7D,F,G), suggesting that Panx1 overexpression appears to promote the formation of immature spines in WT and Panx1-KO neurons.

Together, these results indicate that Panx1 is implicated in the modulation of dendritic arborization and spine maturation.

### 3.4. The lack of Panx1 Promotes F-actin Polymerization in Hippocampal Neurons via the Activation of Rac1 and Repression of the RhoA GTPases

As the actin cytoskeleton regulates the morphology of dendrites and spines, the trafficking of glutamate receptors, and the molecular organization of the PSD [68], we explored whether the morphological changes observed in Panx1-KO neurons were related to actin cytoskeletal remodeling. Actin exists in a dynamic equilibrium between the F-actin and G-actin forms, which are abundant in both pre- and post-synaptic compartments [68]. Therefore, we isolated actin filaments and monomers from hippocampal slices to quantify their relative amounts and the F-actin/G-actin ratio (F/G) as an estimation of actin polymerization. We found that the F/G ratio was significantly higher in Panx1-KO compared to WT slices (Figure 8A; WT versus KO; *** *p* < 0.001 Mann–Whitney test), suggesting that the Panx1 ablation favors actin polymerization. Additionally, we investigated the impact of the Panx1 ablation on the F-actin content in brain slices of WT and Panx1-KO mice, by visualization of the phalloidin fluorescence.

We observed a higher rhodamine-phalloidin reactivity in the brain slices from Panx1-KO compared to WT mice (Figure 8B,C; WT versus KO *** *p* < 0.001; Mann–Whitney test), suggesting that the Panx1 ablation promotes the F-actin assembly. We reasoned that if the Panx1 ablation promotes the formation of F-actin, then regulatory proteins that control actin dynamics and organization could mediate this effect. Thus, we evaluated the expression levels of actin-binding proteins (ABPs) and members of the Rho GTPase family, which are master regulators of the actin cytoskeleton [69]. Western blot analysis revealed an increased expression of Arp3, Drebrin, Cortactin 1, Rac1, Cdc42, and RhoA in hippocampal homogenates from Panx1-KO compared to WT samples (Figure 8D,E; WT versus KO *** *p* < 0.001; two-way ANOVA test), indicating that the Panx1 deficiency affects actin remodeling by altering the expression of ABPs and Rho GTPases.

Rho GTPases are important molecular “switches” that transduce extracellular signals to the actin cytoskeleton [68,69]. RhoA, Rac1, and Cdc42 are members of the Rho GTPases family, implicated in the maintenance and reorganization of dendritic structures [70,71]. Since the Rho GTPase family regulates actin dynamics, we hypothesize that the Panx1 channel may control the expression and activity of the Rho GTPases. We observed significantly increased levels of the active forms of Rac1 in Panx1-KO mice compared to the WT samples and, unexpectedly, we also found a dramatic reduction in the activated form of RhoA in Panx1-KO tissues (Figure 8, F and G; WT versus KO * *p* < 0.005, *** *p* < 0.001; two-way ANOVA test). On the contrary, there were no significant differences in the expression of the activated form of Cdc42.

All these data support a modulatory role of Panx1 for the formation and activity of synapses, which seems to be dependent on the actin cytoskeleton.

## 4. Discussion

Panx1 channels are non-selective channels essential for cellular communication under physiological and pathological conditions [4,72]. At the CNS, they have been proposed as acting as negative modulators of the neuronal activity as the Panx1 deletion, its knockdown, and its pharmacological inhibition enhance the glutamatergic neurotransmission, LTP, and neurite outgrowth [30,31,33,73]. Moreover, the overexpression of Panx1 has been associated with neuronal hyperactivity [74] and death [75], and aberrant activity of Panx1 has been observed in several brain disorders, including ischemia [24,25], epilepsy [26,27], and Alzheimer’s disease [29]. Therefore, the expression and activity of the Panx1 channels appear to be critical for the proper functioning of the brain circuits and especially relevant in the onset of synaptic disorders. Here, we explored the consequences of the constitutive global long-term ablation of Panx1 on the neuronal excitability, excitatory synaptic structure, and function in CA1 pyramidal neurons, as well the impact of the Panx1KO in the actin cytoskeleton organization and neuronal morphology. It is noteworthy that constitutive gene deletion could imply some problems affecting the correct interpretation of the results. For instance, constitutive deletion of Panx genes have been reported as producing some contradictory results between different groups, which can be a consequence of incomplete penetrance of genetic deletion and/or compensatory overexpression of other related proteins [76,77]. However, our previous findings using this particular Panx1-KO mouse, which was characterized by alteration in long-term synaptic plasticity and cognitive function [30,52], have been reproduced using other animal models [31]. Thus, in the present study, we explored if those effects could be due to modifications in the establishment of the neuronal circuits, specifically in the hippocampus.

By using electrophysiological, biochemical, and structural approaches, we demonstrated that, in the absence of Panx1, compensatory mechanisms operate that produce alterations in the Rho GTPases activity and actin cytoskeleton remodeling. Modifications in the actin organization in the Panx1-KO neurons enhanced the dendritic branching and spine maturity, leading to modifications in the number of synaptic contacts in the brain of Panx1-KO mice compared to their WT littermates.

### 4.1. The Ablation of Panx1 Perturbs Neuronal Excitability without Affecting the Spontaneous Release of Glutamate

Under intense neuronal activity, such as during the stimulation to evoke input/output curves, the absence or the blockade of Panx1 induces an increase in the excitatory synaptic transmissions [30,31,73]. However, it is still unknown how Panx1 could affect synaptic transmission under basal conditions. Therefore, we monitored glutamatergic neurotransmission and synaptic connectivity to determine whether the absence of Panx1 elicits adaptations that regulate synaptic strength. We found that CA1 pyramidal neurons from Panx1-KO mice exhibited a lower threshold for triggering APs and fired more APs in response to a depolarizing current ramp, showing higher currents at positive potentials and a lower spike frequency adaptation (Figure 1). We also observed higher input resistance in the Panx1-KO neurons, suggesting that CA1 pyramidal neurons in the Panx1-KO brains are more excitable. In this regard, similar changes in the threshold for AP discharges have been associated with neuronal maturation and homeostatic mechanisms [78,79]. Furthermore, similar homeostatic modifications have been produced by changes in the activity of different ion channels in CA1 neurons [80], including potassium currents I_A_ and I_D_ [81,82,83], hyperpolarization-activated current I_h_ [84], and persistent TTX-sensitive sodium current (I_NAP_) [85]. In addition, the lower spike-firing adaptation observed in Panx1-KO neurons can imply a change in the expression and distribution of the voltage-gated K^+^ channels. In this regard, the Kv3 subfamily of channels limits the duration of the APs and ensures a quick recovery of the voltage-gated Na^+^ channels; the Kv2 subfamily of channels are high-threshold but with slower activation and inactivation kinetics. Whether the absence of Panx1 produces modifications in the Kv3 and Kv2 K^+^ channels, it could certainly modify the repolarizing/after-hyperpolarizing phases and hence regulate the interspike interval and conduction fidelity during sustained stimulation [86,87].

On the other hand, changes in the KCC2 transporter, which regulates neuronal chloride gradients and GABA signaling, also modulates the intrinsic neuronal excitability [88], suggesting that GABA transmission could be an important factor affected in the Panx1-KO neurons. Thus, although inhibitory transmission was not addressed in the present study, modifications in the excitatory/inhibitory (E/I) balance is an aspect that deserves consideration and requires further investigation. In addition to a large pore conformation with high conductance (100–550 pS) that mediates a non-selective ionic flux and ATP release [6,11,17], Panx1 channels also show a constitutive small pore activity characterized by low conductance (50–80 pS) driving a chloride permeability at negative voltages and outwardly rectifying current-voltage relations [7,8,9,10], which could influence the electrical properties of the cells and that can explain the modifications in the AP threshold upon Panx1 ablation. Accordingly, a recent report revealed higher excitability in the hippocampus of Panx1-KO mice [89].

To test if excitability changes involved modifications in excitatory glutamatergic synaptic transmission, we employed electrophysiological recording of evoked (fEPSPs) and spontaneous (sEPSCs, and mEPSCs) synaptic events. We did not find any significant differences either in the initial slope of fEPSPs or afferent FV amplitude, suggesting non-apparent changes in the efficacy of the evoked excitatory synaptic transmission between Panx1-KO and WT neurons. However, AR and NR-FP revealed that Panx1-KO neurons exhibit an enhanced tendency to fire pop-spikes consistent with an impairment in the E/I balance or a modification in the intrinsic excitability of the CA1 neurons. Since GABAergic neurons regulate and tune the activity of pyramidal neurons and modulate the oscillatory activity of neural networks, the inhibitory GABAergic transmission and oscillatory activity upon Panx1 intervention needs to be evaluated in the future.

The latent higher excitability revealed by the changes in the firing threshold and the generation of pop-spikes seem not due to an increase in the spontaneous basal synaptic transmission as the frequency, and the amplitude of the sEPSCs were indistinguishable between groups. Likewise, the frequency of mEPSCs were similar, indicating that spontaneous synaptic activity is normal in Panx1-KO neurons (Figure 1D,F). However, the amplitude of mEPSCs was higher, which could correlate with the higher excitability shown by these neurons. The fact that the spontaneous neurotransmission was unaltered, but the evoked glutamate release seems prompted to respond to activity demand suggest that under Panx1 ablation, CA1 neurons could use a different release molecular machinery, segregated at distinct postsynaptic sites, as has been previously proposed for periferic and central synapses (reviewed in [90,91]). These spontaneous events independent of APs inform about the potential locus for synaptic modification. The amplitude of mEPSCs is thought to reflect changes in the expression (number) or activity (conductance) of ionotropic glutamate receptors; meanwhile, changes in the frequency of mEPSCs has been related to presynaptic mechanisms such as the quantal content and/or the number of synapses [91]. The fact that the amplitude of mEPSCs was higher in Panx1-KO neurons strongly supports postsynaptic modifications, although we cannot exclude that the lack of Panx1 also affects pre-synaptic transmission mechanisms, as we will discuss latter.

In this regard, Bialecki et al. reported that the Panx1 blockade or interfering with the NMDAR-Panx1-Src kinase pathway induces an increase in the sEPSC frequency without modifications in the evoked neurotransmission [73]. The mechanism involved an increase in anandamide levels and the presynaptic TRPV1 activation. Although Bialecki et al. described the same results in conditional Panx1-KO neurons, those effects were seen in younger animals (P21–P37) under short-term Panx1 deletion (Panx1-KO once daily i.p. tamoxifen injections for 5 d) [73,92]. Thus, it is feasible that when we evaluated the synaptic responses in aged animals (6 m.o) a different mechanism could take place in the constitutive global Panx1-KO mice, in which long-term homeostatic mechanisms could occur allowing the maintenance of the synaptic transmission in the adult hippocampus without alterations in the spontaneous neurotransmitter release. Nevertheless, we cannot rule out the contribution of the presynaptic TRPV1-mediated facilitation of glutamate release upon the Panx1-ablation observed by Bialecki et al. in our experimental conditions [73].

### 4.2. The Release Probability and RRP of Synaptic Vesicles Are Increased in Panx1-KO Neurons

In the adult Panx1-KO hippocampus, we previously reported an increased induction of LTP and a lower capacity to induce LTD, which was a potentiation instead of a depression, using standard electrophysiological protocols [30,31]. Despite that, the chemical induction of LTD revealed that Panx1-KO synapses have all the protein machinery to support this plastic modulation [30,31]. Our present findings show that in response to a use-dependent depression during a high-frequency stimulus train, Panx1-KO neurons exhibit a sustained response indicating a greater size of RRP (Figure 2). This observation could explain the potentiated response observed in Panx1-KO slices after applying a protocol that normally induces LTD in WT animals [30,52].

An increase in the glutamate release could result from increasing the number of Ca^2+^-responsive vesicles (i.e., an increase in the RRP), or an increase in the P_r_. The examination of the PPR revealed significant changes at 300 ms of the interstimulus interval between groups. Accordingly, the use-dependent depression induced by a 10 Hz stimulus train showed a reluctance of the Panx1-KO synapses to deplete the RRP vesicles evoked by the high-frequency stimulation. Consistently, cumulative EPSCs were significantly increased at Panx1-KO synapses, suggesting that the pool of synaptic vesicles immediately available for release could be greater in Panx1-KO neurons.

Interestingly, recent evidence highlights the role of Panx1 channels in the homeostatic adjustment of the synaptic strength in hippocampal cultures upon chronic inactivity [43]. Using the immunodetection of the glutamate-transporter 1 (vGlut1) as an index of the presynaptic strength, Rafael et al., 2020, reported that Panx1 channels are required for the compensatory vGlut1 upregulation in the presynaptic terminals and the adjustment of synapse density under chronic inactivity.

### 4.3. Long-Term Panx1 Ablation Affects Structural Connectivity

The present study also showed that the multiplicity index, an estimation of the number of releasing sites or likely synaptic contacts [48,49], was significantly lower in Panx1-KO neurons (Figure 3J). In contrast, the ultrastructural findings revealed a greater number of synaptic contacts in Panx1-KO neurons (Figure 5), so compensatory mechanisms should operate in Panx1-KO synapses to preserve the spontaneous glutamate release and synaptic transmission under resting activity. In this regard, the highly branched dendritic tree of central neurons defines how neurons can receive synaptic inputs from other neurons and strongly influences how these inputs are integrated to allow signal transmission and computation [93]. Our results revealed a more complex morphology of the CA1 neurons from Panx1-KO brains, as disclosed by the intersection profiles obtained by counting the number of dendritic branches at a given distance from the soma. In addition, Panx1-KO neurons displayed longer dendrites with more ramifications in both the apical and basal segments and higher branch order. Indeed, this greater dendritic complexity supposed that Panx1-KO CA1 neurons could possess the morphology, structure, and protein composition for more increased connectivity. In fact, Panx1-KO neurons exhibit a more significant percentage of mature-like dendritic spines, a higher proportion of multiple synaptic contacts, spines with a larger PSD size, and a higher number of synaptic vesicles per bouton or active zone compared to WT neurons (Figure 5D–F). It has been demonstrated that mushroom- and cup-shaped spines are associated with an increase in the size of the PSD and synaptic efficacy [94,95]. Furthermore, the results obtained under depletion experiments indicate that Panx1-KO mice exhibit a greater size of the RRP, as evidenced by the increase in the synaptic responses after the subsequent stimuli. In addition, similar relationships between the number of vesicles in the docked vesicle pool, the size of the RRP, the active zone, and the PSD sizes in hippocampal synapses have been previously reported [96]. Moreover, PSD- and SM-enriched fractions obtained from hippocampal slices disclosed an enhanced expression of pre- and post-synaptic proteins (Figure 5H), suggesting that the Panx1 ablation might induce compensatory modifications in both compartments.

The morphological changes in dendritic arborization and spines, along with a change in the synaptic protein composition, likely reflect a modification in the remodeling of the neuronal cytoskeleton. We tested this hypothesis by evaluating the organization of the actin cytoskeleton. Actin cytoskeleton transits in a dynamic equilibrium between F-actin and G-actin forms, abundant in presynaptic terminals and postsynaptic dendritic spines [68]. Furthermore, this equilibrium between G-actin and F-actin is finely and rapidly regulated during activity by many postsynaptic ABPs. In this study, we reported that Panx1-KO brain tissue exhibited a higher content of F-actin, suggesting that the Panx1 ablation promoted a disequilibrium in actin dynamics towards polymerization. Although these findings revealed by phalloidin fluorescence and F/G ratio were evident in Panx1 tissue, further confirmation of actin cytoskeleton dynamics will require more resolutive techniques such as two-photon or superresolution microscopy using monomeric and filamentous actin dyes.

In support of a potential regulatory role of Panx1 on the actin cytoskeleton, it has been previously reported that Panx1 channels influence cellular changes that require cytoskeletal modifications, including migration, differentiation, and proliferation [33,97]. Interestingly, Panx1 directly interacts with F-actin and its regulator Arp3 [33,98], which is the main component of the Arp2/3 complex involved in the nucleation and branching of F-actin [99]; thus, suggesting that F-actin polymerization and nucleation of new microfilaments might directly control Panx1 localization. In this context, a higher spine density and complexity network has been reported in global and conditional Panx1-KO [32] compared to control animals, suggesting that Panx1 activity limits the neuronal actin cytoskeleton remodeling. The findings by Sanchez-Arias et al. 2019 were performed in primary cortical neuron cultures (DIV 12–14) obtained from P0 or cortical neurons from young P14 and P29 mice, compared to our results in 6 m.o animals. Since the brain has the highest expression of Panx1 in neurons and glial cells at the early stages of development [30,60,61,100], the expression of impaired Panx1 channels may influence neuronal development. However, neurons could not be sufficient to account for the structural plasticity, considering that glial cells provide the proper environment for neuron maturation and synapse formation, pruning, and plasticity [101,102]. Increasing evidence suggests bi-directional interactions between neurons and astrocytes at excitatory synapses, which modulate synaptic strength [103]. However, in the present study, we did not test the direct impact of the glial Panx1 on neuronal morphology and future studies will be necessary to describe their contribution to this process.

One of the critical aspects of establishing and maintaining the neuronal structure is the well-controlled turnover of the cytoskeletal elements [104]. At this point, Rho GTPases are the main regulators that control the organization and dynamics of the actin/tubulin cytoskeleton [69]. Among them, Rac1, Cdc42, and RhoA play a major role in dendritic spine dynamics, connecting signals from the postsynaptic receptors to changes in ABPs and hence, actin polymerization/depolymerization [70,71]. While Rac1 and Cdc42 activation promote spine formation and stabilization, RhoA activation leads to spine retraction and pruning [105,106,107]. Accordingly, we found that hippocampal tissue from Panx1-KO mice exhibited enhanced Rac1, Cdc42, and RhoA levels along with selective ABPs, including Arp3, Drebrin, and Wave-1. Interestingly, the active form of Rac1, but not Cdc42, was significantly increased in hippocampal homogenates from Panx1-KO mice. In contrast, the active form of RhoA was almost absent, consistent with their antagonistic role in dendritic and spine morphology [105]. Since Panx1 interacts with actin and Arp3, this association could be relevant to actin-dependent processes [33]. At this point, it has been described that downstream effectors of Cdc42 and Rac1, such as N-WASP and Arp2/3 complex, promote the reorganization of the actin cytoskeleton leading to rapid stabilization of dendritic spines [108]. In fact, the Arp2/3 complex is regulated by Rho GTPases such as Rac1 or Cdc42, which stimulate the complex via downstream effectors such as WAVE [109]. On the other hand, RhoA can activate mDia1, an actin nucleator, to drive Arp2/3-independent formation of stress fibers, while RhoA effectors, such as ROCK and formins mDia3, promote actin depolymerization and spine instability [110,111]. This evidence suggests that the Rho GTPase signaling could be involved in the morphological effects observed in Panx1-KO mice. In this regard, a non-canonical RhoA-mDia-HDAC6 signaling pathway was recently described for α1AR activation of Panx1 channels [112] according to the previously reported reduction in ATP release via Panx1 channels after RhoA inhibition [113]. Evidence suggests the involvement of RhoA in the activation of Panx1. First, the inhibition of RhoA disrupts the actin cytoskeleton, significantly reducing ATP release via mechanosensitive Panx1 channels [113]. Second, the inhibition or knockdown of Rho GTPases, or the disruption of the cytoskeleton with nocodazole or cytochalasin D, reduces ATP release via Panx1 in Schwann cells [114].

It is also worth mentioning that RhoA and Rac1/Cdc42 typically exhibit an antagonistic relationship to control protein processing and trafficking as well as synaptic remodeling [105,115,116,117]. Consistent with these previous studies, we observed a reduced activation of RhoA and an increased Rac1 activation. Interestingly, Rac1/Cdc42 activation cooperates with increased F-actin content and neuronal morphology changes in absence of Panx1. The mechanisms by which Panx1 promotes Rho activation/inactivation remain to be elucidated. Unsurprisingly, there is a feedback mechanism between Panx1 and the actin cytoskeleton. Thus, it appears that the signaling-cascades mediated by Panx1 channels control the steady-state activity levels of the Rho GTPases Rac1 and RhoA, but the mechanism by which Rac1/RhoA impacts on the actin cytoskeleton dynamics upon Panx1 absence remains to be elucidated.

These findings revealed an imbalance in the equilibrium towards more mature spines, driven by an enhanced actin polymerization in Panx1-KO neurons, implying a more static than plastic synaptic architecture produced by constitutive Panx1 ablation. These observations can also explain the previously reported difficulty of inducing LTD mechanisms which implies a retraction or reduction in the size and number of dendritic spines and a higher depolymerization of actin filaments. Consistently, despite the fact that Panx1 ablation has been associated with enhanced glutamatergic neurotransmission, LTP, and neurite outgrowth [30,31,33], alterations in learning flexibility, such as those implicated in the novel object recognition memory, the reversion of the spatial and working memory, have been described in Panx1-KO mice [31,52], which involves the weakening or elimination of synapses to allow the storage of traces of new memories [118,119,120]. Nevertheless, these animals still preserve their ability to learn, suggesting that compensatory or homeostatic mechanisms must operate to maintain the plasticity of the adult brain at cognitively significant levels. The synaptic location of Panx1 and/or their signaling could be part of a molecular machinery that recruits actin cytoskeleton to regulate bidirectionally changes in synaptic strength. Therefore, Panx1 expression and activity in the brain seem to be critical to the proper functioning of the neural circuits.

Overall, the results presented here are consistent with a critical role of Panx1 channels for the structure and functioning of hippocampal CA1 neurons. Interestingly, the present data also provide evidence of a possible link between Panx1 and Rho GTPase-dependent regulation of the actin cytoskeleton. Thus, the functional interaction between Panx1 channels and Rho GTPases could be of pivotal relevance considering that numerous brain disorders such as neuropsychiatric and neurodegenerative conditions are manifested with abnormalities in the neuronal actin cytoskeleton.

## 5. Conclusions

The permeation of ions and metabolites throughout Panx1 channels modulates neuronal excitability and circuit formation in the mammalian brain. Here, we show that the absence of Panx1 leads to compensatory structural and functional modifications in the hippocampal pyramidal neurons of the Panx1-KO mice, enhancing neuronal excitability. Furthermore, these changes correlate with an increase in the complexity of the dendritic arborization, the presence of multiple synaptic contacts, and a higher proportion of mature dendritic spines, all of which seem to rely on enhanced F-actin polymerization. This latter appears to be caused by an imbalance in the Rac1/RhoA activity, suggesting a novel interplay between Panx1 and the small Rho-GTPases that could modulate the actin cytoskeleton-remodeling, controlling neuronal morphology and functionality.

## Figures and Tables

**Figure 1 cells-11-03646-f001:**
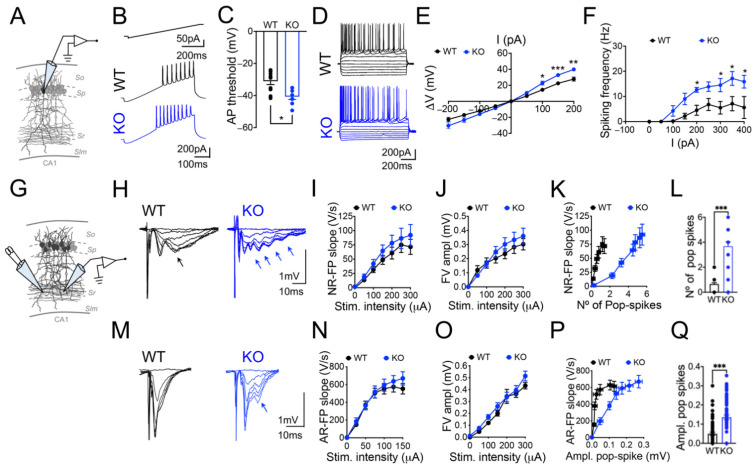
Enhanced excitability but normal glutamatergic synaptic transmission in hippocampal CA1 neurons from Panx1-KO mice. (**A**) Schematic drawing of a whole cell recording in the pyramidal cell layer (*Stratum pyramidale, Sp*) of the CA1 region of a hippocampal slice. (**B**) Representative traces of action potential recordings evoked by a current ramp in CA1 neurons of WT (black) and Panx1-KO (KO, blue) mice. (**C**) Action potential threshold. *n* = 10 (WT) and *n* = 8 (KO) cells from 4–5 animals, * *p* = 0.0108 Mann–Whitney test. (**D**) Representative traces of membrane potential changes in response to the current steps. (**E**) Current-voltage curves. *n* = 10 (WT) and *n* = 8 (KO) cells from 4–5 animals, * *p* < 0.0142, ** *p* < 0.046 and *** *p* < 0.001 two-way ANOVA test. (**F**) Spiking frequency in response to the current steps for WT and KO hippocampal cells. *n* = 10 (WT) and *n* = 8 (KO) cells from 4–5 animals, * *p* < 0.0142 two-way ANOVA test. (**G**) Schematic drawing of a field recording in the dendritic cell layer (*Stratum radiatum, Sr*) of the CA1 region of a hippocampal slice. (**H**) Representative traces of input–output curves of pharmacologically isolated NMDAR fEPSP (NR-FP) and analysis of the slope (**I**), fiber volley (FV) amplitude (**J**), plots of NR-FP slope versus the number of pop spikes (**K**), and averaged number (N) of pop spikes (**L**). *n* = 10 (WT) and *n* = 7 (KO) cells from 4–5 animals, *** *p* < 0.001 Mann–Whitney test. (**M**) Representative traces of input–output curves of AMPAR fEPSP (AR-FP) and analysis of the slope (**N**), FV amplitude (**O**), plots of AR-FP slope versus the amplitude of the pop spike (**P**), and averaged amplitude of pop spikes (**Q**). *n* = 22 (WT) and *n* = 25 (KO) cells from 5–6 animals, *** *p* < 0.001 Mann–Whitney test.

**Figure 2 cells-11-03646-f002:**
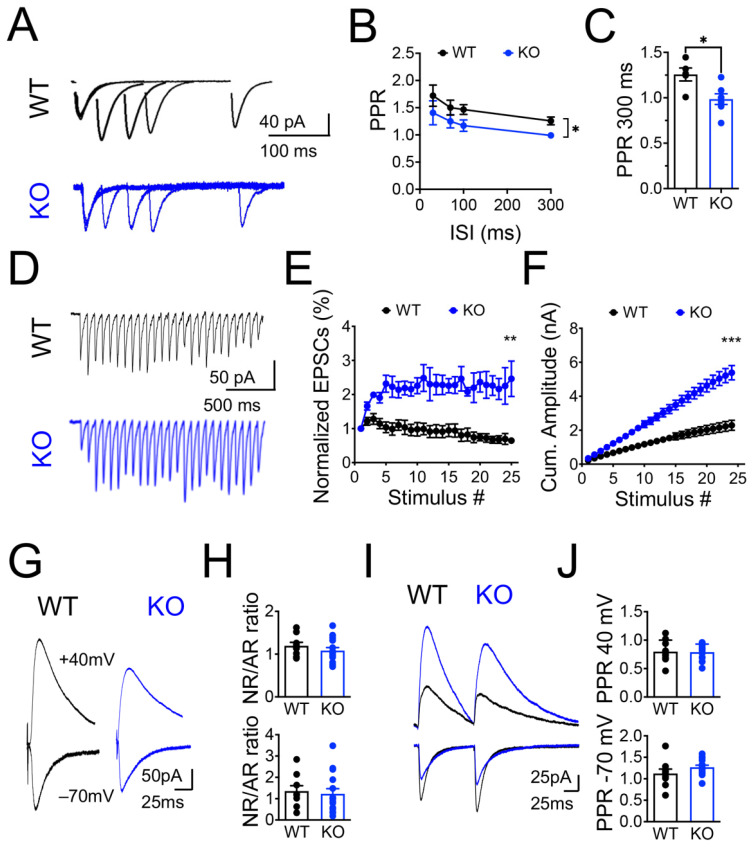
Readily releasable pool (RRP) and vesicle release probability are increased in Panx1-KO synapses. (**A**) Representative traces of superimposed paired-pulse responses at variable inter-stimulus intervals (ISI) from WT (black) and Panx1-KO (KO, blue) CA1 neurons. (**B**) Analysis of the paired-pulse facilitation ratio (PPR). *n* = 5 (WT) and *n* = 7 (KO) cells from 4–5 animals, * *p* = 0.0074 two-way ANOVA test. (**C**) PPR at 300 ms ISI. * *p* = 0.0144 Mann–Whitney test. (**D**) Representative traces of EPSCs were evoked by a train of 25 pulses at 10 Hz. (**E**) Normalized values of EPSCs. *n* = 5 (WT) and *n* = 7 (KO) slices from 4–5 animals, ** *p* = 0.0017 two-way ANOVA test. (**F**) Plot of the cumulative EPSCs versus number of stimuli. *n* = 5 (WT) and *n* = 7 (KO) slices from 4–5 animals, *** *p* =0.002 two-way ANOVA test. (**G**) Representative traces of the NMDAR and AMPAR currents. (**H**) Analysis of AMPAR to NMDAR (AR/NR) ratio recorded at 40 mV (**top**) and −70 mV (**bottom**). (**I**) Representative traces of AMPAR and NMDAR currents induced by a paired-pulse (at 50-ms intervals). (**J**) Analysis of paired-pulse ratio (PPR) recorded at 40 mV (top) and –70 mV (**bottom**).

**Figure 3 cells-11-03646-f003:**
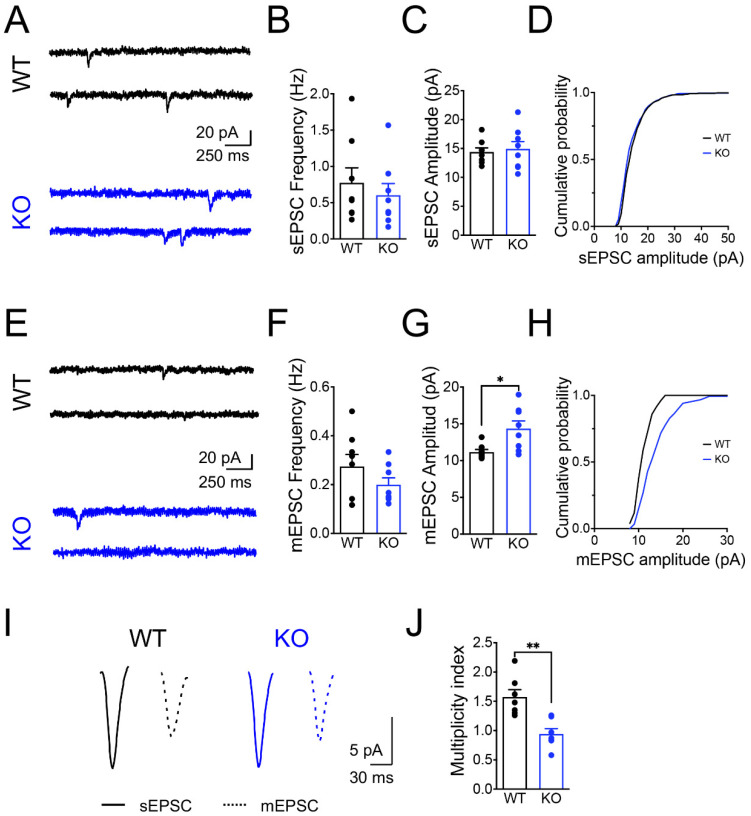
Increased mEPSC amplitude but normal spontaneous release and reduced number of releasing sites in Panx1-KO CA1 neurons. (**A**) Representative traces of sEPSC events recorded in WT (black) and Panx1-KO (KO, blue) CA1 neurons. *n* = 8 (WT) and *n* = 8 (KO) cells from 4–5 animals. (**B**,**C**) Analysis of sEPSC frequency (**B**) and amplitude (**C**). (**D**) Cumulative probability plots of the sEPSC amplitude distribution. (**E**) Representative traces of mEPSC events. (**F**–**H**) Analysis of mEPSC frequency (**F**) and amplitude (**G**). *n* = 8 (WT) and *n* = 8 (KO) cells from 4–5 animals, * *p* = 0.0104 Mann–Whitney test. (**H**) Cumulative probability plots of the mEPSC amplitude distribution. (**I**) Averaged traces of individual sEPSC (continuous line) and mEPSC (dotted line) events. (**J**) Analysis of the multiplicity index. *n* = 8 (WT) and *n* = 8 (KO) cells from 4–5 animals, ** *p* = 0.012 Mann–Whitney test.

**Figure 4 cells-11-03646-f004:**
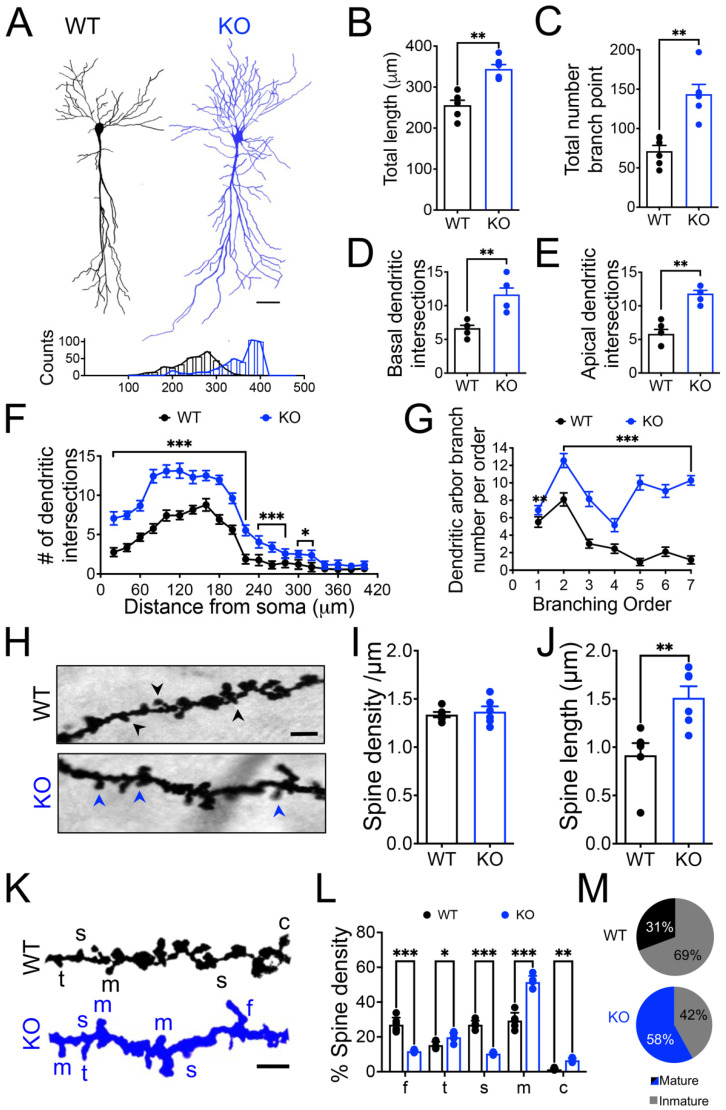
Enhanced dendritic arborization and spine maturation in Panx1-KO neurons. (**A**) Representative drawings of the Golgi-stained CA1 neurons (**top**) and histogram distribution of total dendritic length (**bottom**) for WT (black) and Panx1-KO (KO, blue) mice. Magnification, 40X; bar: 40 µm. (**B**) Averaged total dendritic length. *n* = 100 (WT) and *n* = 100 (KO) neurons from 6 animals, ** *p* = 0.002 Mann–Whitney test. (**C**) Averaged basal dendritic branches. *n* = 100 (WT) and *n* = 100 (KO) neurons from 6 animals, ** *p* = 0.002 Mann–Whitney test. (**D**) Averaged apical dendritic branches. *n* = 100 (WT) and *n* = 100 (KO) neurons from 6 animals, ** *p* = 0.002 Mann–Whitney test. Dendritic branches. (**E**) Averaged dendritic intersections. *n* = 100 (WT) and *n* = 100 (KO) neurons from 6 animals, ** *p* = 0.002 Mann–Whitney test. Dendritic branches. (**F**) Number of intersections as a function of the distance from soma. *n* = 100 (WT) and *n* = 100 (KO) neurons from 6 animals, * *p* = 0.02, *** *p* < 0.001 two-way ANOVA test. (**G**) Branch order as a function of the distance from soma. *n* = 100 (WT) and *n* = 100 (KO) neurons from 6 animals, *** *p* < 0.001 two-way ANOVA test. (**H**) Representative images of dendritic segments with dendritic spines (arrowheads). (**I**) Averaged spine density. (**J**) Averaged spine length. *n* = 80 (WT) and *n* = 80 (KO) dendrites from 6 animals, ** *p* = 0.004 Mann–Whitney test. (**K**) Pseudo-colored images of dendritic segments as in (**H**) showing different types of dendritic spines, filopodium (f), thin (t), short (s), and mushroom (m) types. Magnification 100X, bar: 2 µm. (**L**) Proportion of different types of dendritic spines. (**M**) Percentage of mature and inmature dendritic spines. *n* = 80 (WT) and *n* = 80 (KO) dendrites from 6 animals, * *p* = 0.02, ** *p* = 0.005, *** *p* < 0.001 two-way ANOVA test.

**Figure 5 cells-11-03646-f005:**
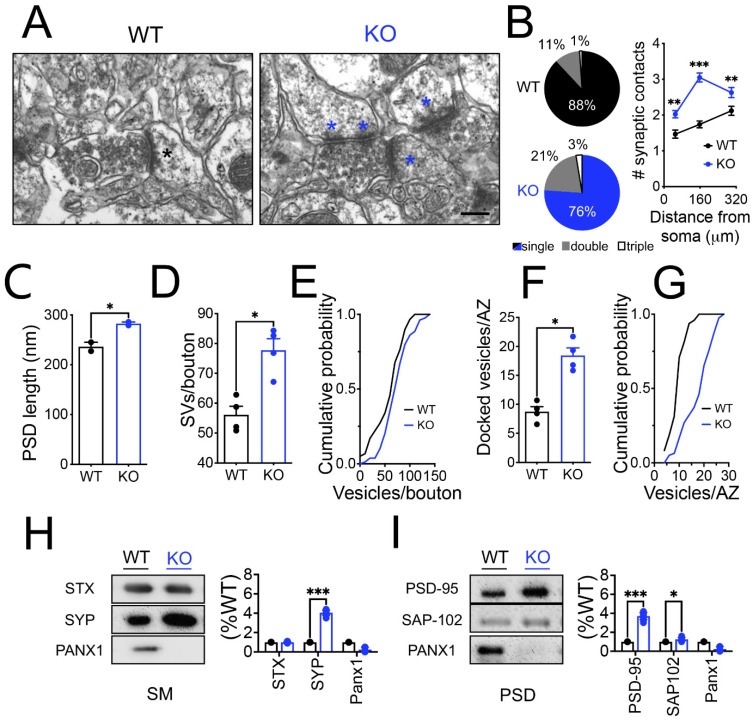
Multiple contacts, a higher number of docked vesicles, and enhanced PSD length in Panx1-KO synapses. (**A**) Representative transmission electron microscopy photographs of assymetric synapses of the CA1 *Stratum radiatum* area of WT (black) and Panx1-KO (KO, blue) mice. Magnification 43.000X, bar: 200 nm. (**B**) Percentage of single, double, and triple contacts. *n* = 276 (WT) and *n* = 392 (KO) synaptic contacts, 6 ultrathin sections from 3 animals, ** *p* = 0.002, *** *p* < 0.001 two-way ANOVA test. (**C**) PSD length. * *p* = 0.03 Mann–Whitney test. (**D**) Number of synaptic vesicles per bouton. *n* = 59 (WT) and *n* = 79 (KO) synaptic boutons, 6 ultrathin sections from 3 animals, * *p* = 0.0286 Mann–Whitney test. (**E**) Cumulative probability of the docked vesicle distribution. (**F**) Number of docked vesicles at the active zone (AZ). *n* = 48 (WT) and *n* = 78 (KO) synaptic boutons, 6 ultrathin sections from 3 animals, * *p* = 0.03 Mann–Whitney test. (**G**) Cumulative probability of the vesicles/AZ distribution. (**H**) Representative blots and densitometric analysis of synaptic proteins membranes avoided of PSD (SM)-enriched fractions. *n* = 6 (WT) and *n* = 6 (KO) ultrathin sections from 5–6 animals, *** *p* < 0.001 two-way ANOVA test. (**I**) Representative blots and densitometric analysis of synaptic protein levels in synaptic and PSD-enriched. *n* = 6 (WT) and *n* = 6 (KO) ultrathin sections from 5–6 animals, * *p* = 0.002, *** *p* < 0.001 two-way ANOVA test.

**Figure 6 cells-11-03646-f006:**
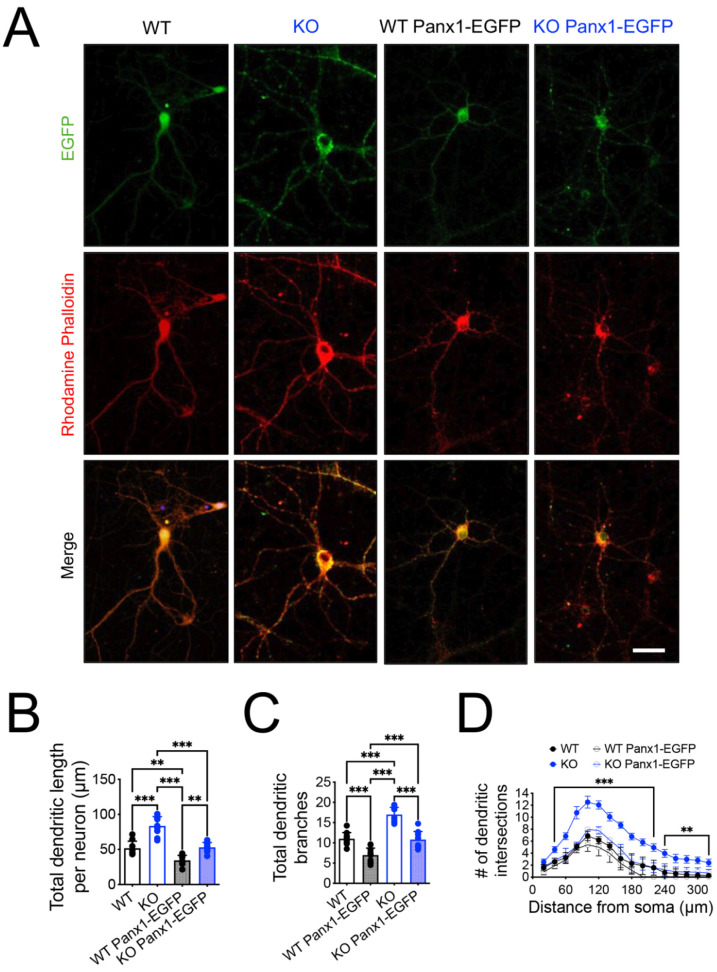
Panx1 regulates morphology in cultured hippocampal neurons. (**A**) Representative images of DIV 14 hippocampal neurons transfected with EGFP or Panx1-EGFP (**green**) and stained with rhodamine-phalloidin (**red**). Magnification, 40X; Scale bar, 15 µm. *n* = 10 (WT) and *n* = 14 (KO) cells from 5–6 cultures. (**B**) Quantification of dendritic length. *n* = 54 (WT) and *n* = 55 (KO) cells from 5–6 cultures, ** *p* = 0.003, *** *p* < 0.001 two-way ANOVA test. (**C**) Total of the dendritic branch. *n* = 54 (WT) and *n* = 55 (KO) cells from 5–6 cultures, *** *p* < 0.001 two-way ANOVA test. (**D**) Sholl analysis of dendritic arbors. *n* = 54 (WT) and *n* = 55 (KO) cells from 5–6 cultures, ** *p* < 0.005, *** *p* < 0.001 two-way ANOVA test.

**Figure 7 cells-11-03646-f007:**
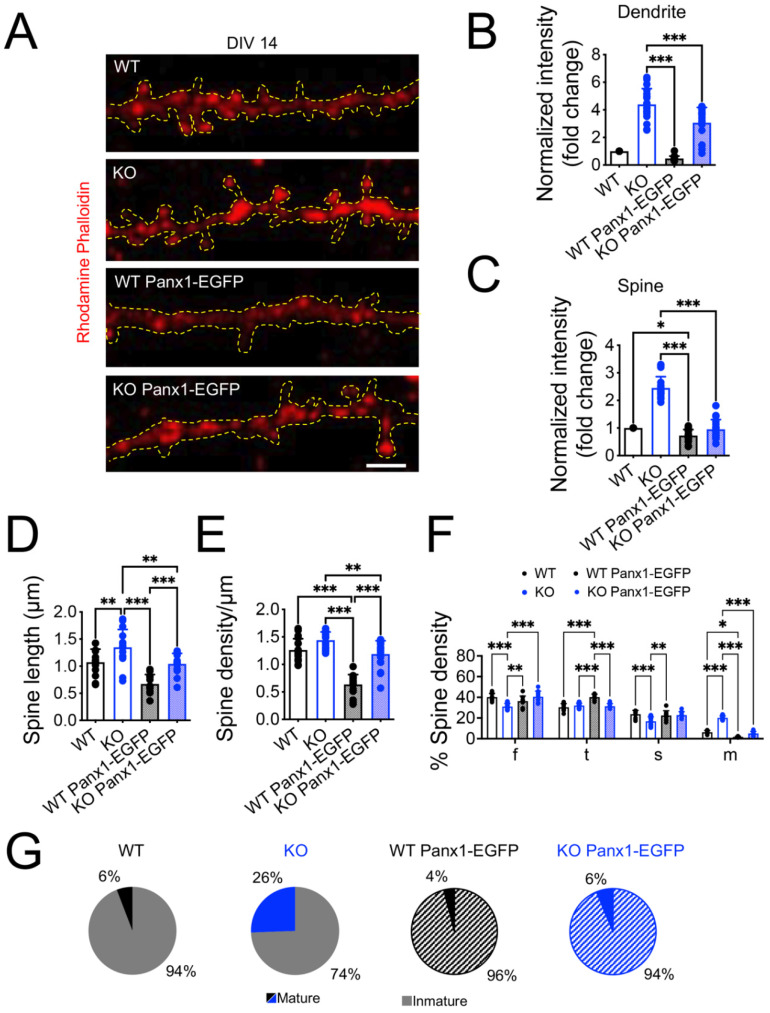
Panx1 cellular and dendritic spine (**A**) Representative dendritic segments of DIV 14 hippocampal neurons transfected with EGFP or Panx1-EGFP (green) and stained with rhodamine-phalloidin (red). Magnification, 100X; Scale bar, 2 µm. (**B**) Quantification of rhodamine-phalloidin intensity in the dendritic shaft. *n* = 23 (WT) and *n* = 28 (KO) dendrites from 5–6 cultures, *** *p* < 0.001 two-way ANOVA test. (**C**) Quantification of rhodamine-phalloidin intensity in dendritic spines. *n* = 23 (WT) and *n* = 28 (KO) dendrites from 5–6 cultures, * *p* = 0.02, *** *p* < 0.001 two-way ANOVA test. (**D**) Quantitative analysis of dendritic spine length. *n* = 23 (WT) and *n* = 28 (KO) dendrites from 5–6 cultures, ** *p* = 0.002, *** *p* < 0.001 two-way ANOVA test. (**E**) Quantitative analysis of spine density. *n* = 23 (WT) and *n* = 28 (KO) dendrites from 5–6 cultures, ** *p* = 0.03, *** *p* < 0.001 two-way ANOVA test. (**F**) Spine classification and quantification of dendritic spines. *n* = 164 (WT) and *n* = 163 (KO) spines from 5–6 cultures, * *p* = 0.01, ** *p* = 0.003, *** *p* < 0.001 two-way ANOVA test. (**G**) Percentage of mature and immature dendritic spines. *n* = 23 (WT) and *n* = 28 (KO) dendrites from 5–6 cultures, *** *p* < 0.001 two-way ANOVA test.

**Figure 8 cells-11-03646-f008:**
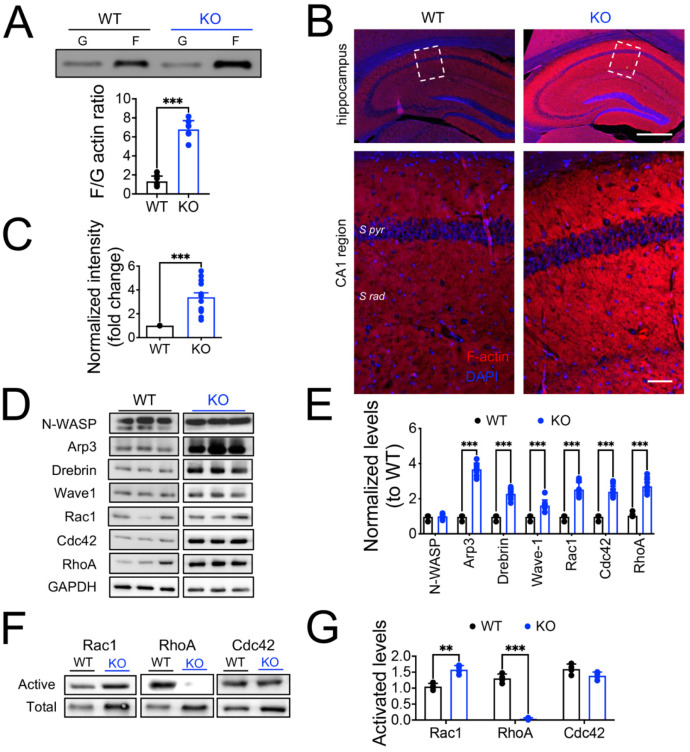
Increased actin polymerization and Rac1 activity in Panx1-KO hippocampi. (**A**) Representative blots (**top**) and densitometric analysis of the relative amount of monomers (**G**) and filaments (F) of actin (**bottom**), in hippocampal lysates of WT (**black**) and Panx1-KO (KO, **blue**) mice. *n* = 6 (WT) and *n* = 6 (KO) slices from 4–5 animals, *** *p* < 0.001 Mann–Whitney test. (**B**) Representative confocal micrographs showing the rhodamine-phalloidin reactivity of the F-actin network in the CA1 region in the hippocampus. Whole hippocampal at 4X magnification (**top panel**), scale bar: 150 µm. Magnified view showing the *Stratum radiatum* (*Sr*) and the *Stratum pyramidale* (*Sp*) layer at 20X magnification (**bottom panel**), scale bar: 25 µm. (**C**) Quantification of rhodamine-phalloidin intensity. *n* = 6 (WT) and *n* = 6 (KO) slices from 6 animals, *** *p* < 0.001 Mann–Whitney test. (**D**,**E**) Representative blots (**D**) and densitometric analysis (**E**) of Rho GTPases and synaptic actin-binding proteins levels. *n* = 6 (WT) and *n* = 6 (KO) slices from 6 animals, *** *p* < 0.001 two-way ANOVA test. (**F**,**G**) Representative blots (**F**) and densitometric analysis (**G**) of the relative levels of the active Rac1, Cdc42 and RhoA proteins. *n* = 6 (WT) and *n* = 6 (KO) slices from 6 animals, ** *p* = 0.02, *** *p* < 0.001 two-way ANOVA test.

**Table 1 cells-11-03646-t001:** Intrinsic passive properties of hippocampal neurons.

	WT	KO	Mann–Whitney Test
	*(n = 10)*	*(n = 8)*	*p-value*
V_m_ (mV)	−71.30 ± 2.83	−68.11 ± 1.42	0.9655
R_in_ (MΩ)	124.64 ± 10.14	170.15 ± 16.90	* 0.0248
C_m_ (pF)	81.09 ± 8.12	77.09 ± 9.16	0.8286
τ	9.83 ± 0.97	12.92 ± 1.65	0.1011
Number of AP	5.44 ± 1.18	8.75 ± 1.30	0.0779
Spiking frequency_200pA_	4.9 ± 2.24	12.62 ± 0.96	* 0.0328
Spiking frequency_250pA_	6.4 ± 2.19	14 ± 1.82	* 0.0204
Spiking frequency_300pA_	5.5 ± 2.23	14.5 ± 2.63	* 0.0247

Input resistance (R_in_), membrane capacitance (C_m_), resting membrane potential (V_m_) and τ were measured in the current clamp on the same cells used to construct I–V curves and determine threshold values.

## Data Availability

The data that support the findings of this study are available from the corresponding author upon reasonable request.

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
