# Peer review of "The Long-Term Pannexin 1 Ablation Produces Structural and Functional Modifications in Hippocampal Neurons"

_cells, 2022, doi:10.3390/cells11223646_

Round 1
Reviewer 1 Report
The manuscript by Flores-Munoz et al. reports on “Long-term Pannexin 1 ablation produces an imbalance between Rho GTPases activity and actin polymerization leading to structural and functional modifications in hippocampal neurons.” The manuscript is well written with new evidence explaining how Panx1 channels (and loss thereof), contributes to synapse function. The authors show convincing evidence that loss of Panx1 causes structural and function modifications at synapse level. Key findings are the novel interaction between Panx1 channels, actin, and Rho GTPases, which are presented as relevant for the synapse stability.
The manuscript is very interesting and well written. The introduction is comprehensive, and the Materials and Methods are described in detail. The results are supported by statistically robust analysis. I do agree with the conclusions made. Overall, this is a mature manuscript and my critic is minor.
Fig.7B. Is there a more convincing way to present phalloidin staining? When slices are made, they never have the same sickness across the entire slice causing artifacts like the edge effects seen here. Is there a way to choose a blue label which is visible?
Use Panx1-KO or any other form you prefer consistently. In lines 513, 697 and elsewhere: Panx1-knockout.
The authors chose to suppress glial cells in their cell culture work. Please elaborate why they compare the role of Panx1 in slices in a scenario with neurons and glial cells and move to a neuron only scenario in vitro. Isn’t this counterintuitive, considering potential roles of glial cells in synapse maturation and function? Isn’t this a way to lose relevant information? See arguments made in lines 908ff.
Considering the amount of work it would take I am only suggesting that the authors should test their key findings by pharmacological blocking of Panx1 using 10Panx1 or Probenecid. In case the authors have already explored this direction, for example by doing Golgi-Cox staining, it would add substantial weight to their data set.
Lines 807ff. In your line of argument add reference 61 to 88 since it shows that Panx1 is localized in a prominent position outside of the PSD in hippocampal neurons. Reference 88 informs about the potential mechanism.
Lines 898ff I would suggest revisiting the evidence provided by the cited research and perhaps rephrase the statement. ß-actin is frequently a contamination in CoIP experiments. The Arp2/3 data are more solid.
Lines 941ff The authors start to dive in this paragraph into an important discussions. Due to potential impact of this paragraph moving the field forward, I would like to see them elaborating with more clarity. Connect the dots going beyond just naming them.
Some images in the Original Images file appear black on black. Is it possible to reverse color and adjust brightness.
Author Response
Reviewer 1
Comments and Suggestions for Authors
The manuscript by Flores-Munoz et al. reports on “Long-term Pannexin 1 ablation produces an imbalance between Rho GTPases activity and actin polymerization leading to structural and functional modifications in hippocampal neurons.” The manuscript is well written with new evidence explaining how Panx1 channels (and loss thereof), contributes to synapse function. The authors show convincing evidence that loss of Panx1 causes structural and function modifications at synapse level. Key findings are the novel interaction between Panx1 channels, actin, and Rho GTPases, which are presented as relevant for the synapse stability.
The manuscript is very interesting and well written. The introduction is comprehensive, and the Materials and Methods are described in detail. The results are supported by statistically robust analysis. I do agree with the conclusions made. Overall, this is a mature manuscript and my critic is minor.
Answer: Thanks to the reviewer for their positive reception of our manuscript. We also appreciate the thoughtful suggestions. As follows, we provide a detailed response to these comments.
Fig.7B. Is there a more convincing way to present phalloidin staining? When slices are made, they never have the same sickness across the entire slice causing artifacts like the edge effects seen here. Is there a way to choose a blue label which is visible?
Answer: We agree with the reviewer's point that the tissue processing implies variability that could lead to misleading interpretations. However, it is important to note that we showed the phalloidin staining analysis obtained from different animals for each strain, finding consistent results. Additionally, we performed the same approach using a GFP-phalloidin getting similar results. Now, we included an additional supplementary figure (figure S3), showing separated 4X photographs of the same slice, allowing the visualization of DAPI and phalloidin staining.
Use Panx1-KO or any other form you prefer consistently. In lines 513, 697 and elsewhere: Panx1-knockout.
Answer: We’ve corrected it with the proper abbreviation in the revised version of the manuscript.
The authors chose to suppress glial cells in their cell culture work. Please elaborate why they compare the role of Panx1 in slices in a scenario with neurons and glial cells and move to a neuron only scenario in vitro. Isn’t this counterintuitive, considering potential roles of glial cells in synapse maturation and function? Isn’t this a way to lose relevant information? See arguments made in lines 908ff.
Answer: The reviewer is right; we cannot exclude the involvement of glial cells in our research design, which was focused on the contribution of Panx1 ablation on the structural and functional plasticity in hippocampal neurons. Therefore, we used dissociated primary neuronal cultures derived from the postnatal brain, which contain a higher number of proliferative glial cells. Furthermore, we clarified that the experimental conditions of our cell culture protocol do not eliminate the glial cells; rather this treatment limits the glial content simplifying the observation of neurons in the culture. See page 8, lines 383-387 in section 2.11 of the material and methods.
We also added an explanation for using primary neurons on page 18, lines 639-643 in the new version of the revised manuscript. Dissociated primary neuronal cultures represent a tool to recapitulate in vitro, many aspects of the brain development that occur in vivo such as neuronal maturation, structural plasticity, and functional activity. We used dissociated primary neuronal cultures derived from the postnatal brain which contain a significant number of glial cells, maintaining their supportive role for neurons.
Considering the amount of work it would take I am only suggesting that the authors should test their key findings by pharmacological blocking of Panx1 using 10Panx1 or Probenecid. In case the authors have already explored this direction, for example by doing Golgi-Cox staining, it would add substantial weight to their data set.
Answer: We agree with the reviewer in this point; further experiments comparing the impact of the Panx1 deletion with its inhibition would be helpful. We will consider the use of Panx1 blockers in our future works as a complement for conditional cell-specific deletion of the Panx1 gene, which is our next step in this research. Unfortunately, 10Panx or Probenecid are not selective drugs. Hence our efforts are now on developing a more specific and cell-type approach to target Panx1.
Lines 807ff. In your line of argument add references 61 to 88 since it shows that Panx1 is localized in a prominent position outside of the PSD in hippocampal neurons. Reference 88 informs about the potential mechanism.
Answer: This argument has been rewritten, and we added some references supporting that (page 26, lines 832-836). However, some references mentioned by the reviewer (61 to 88) not refer to an outer position of Panx1 in the PSD.
Lines 898ff I would suggest revisiting the evidence provided by the cited research and perhaps rephrase the statement. ß-actin is frequently a contamination in CoIP experiments. The Arp2/3 data are more solid.
Answer: Suggestion accepted. This sentence has been rephrased. Page 28, lines 925-932.
Lines 941ff The authors start to dive in this paragraph into an important discussions. Due to potential impact of this paragraph moving the field forward, I would like to see them elaborating with more clarity. Connect the dots going beyond just naming them.
Answer: This argument has been rephrased, and we added some references supporting it (page 29, lines 985-1001).
Some images in the Original Images file appear black on black. Is it possible to reverse color and adjust brightness.
Answer: The reviewer is right; we have changed the figure including the inverted images, in gray scale.
Reviewer 2 Report
The study by Flores-Munoz et al. involves the analysis of Panx1 knock out mice at the adult stage. They combined various electrophysiological assays and morphological analysis of the brain to demonstrate that the absence of Panx1 induces structural and functional alterations in the brain, in particular in the hippocampal neurons used as the major model system during these investigations. They propose that these alterations are related to changes in actin dynamics, and result in an imbalance between members of the Rho GTPase family, and it appears to be the main explanation for the changes in synapse structure and stability reported in this work. Although the lack of Panx1 does not cause lethality, the overexpression of it is likely to contribute to neuronal pathologies, therefore the study is potentially an interest for a broad field of researchers. The analysis is complex and thorough, it involves the application of numerous techniques, the figures are carefully done, clarity of the ms is fine, and I believe that it is suitable manuscript for publication. Nevertheless, it remains a rather descriptive work, and I would highly appreciate more efforts towards mechanistic understanding. For example, what could be the basis of the differential effect on Rac1, cdc42 and RhoA? It is far from being trivial how this gap junction protein impacts on the expression level and activity of the Rho GTPases. An important experiment would be to investigate whether the observed defects and structural alterations are congenital or they develop with age and usage. A developmental analysis of at least some critical phenotypic components would make the paper much stronger.
Minor issues:
Although the use of English is fine in most parts of the paper, several sentences should be modified:
line 748: “operate compensatory mechanisms” was perhaps meant to be “compensatory mechanisms operate”.
line 800: “regulate tune” makes no sense
lines 819 and 831: “cannot discard” should be “cannot exclude”.
line 900: “the Arp2/3 complex involved in de novo nucleation”. The Arp2/3 complex, by definition, is not a de novo nucleation factor as it only works in the presence of an existing filament.
line 914: “in the actin dynamic process” makes no sense.
lines 916-918: “ Activation of Rho GTPases and interaction with numerous ABPs and microtubule-associated proteins extensively interact and give feedback controlling their organization [103,104].” This sentence makes no sense.
line 955: “for proper structure and functioning of neural in hippocampal CA1 neurons”, this sentence makes no sense in this form.
Author Response
Reviewer 2
Comments and Suggestions for Authors
The study by Flores-Munoz et al. involves the analysis of Panx1 knock out mice at the adult stage. They combined various electrophysiological assays and morphological analysis of the brain to demonstrate that the absence of Panx1 induces structural and functional alterations in the brain, in particular in the hippocampal neurons used as the major model system during these investigations. They propose that these alterations are related to changes in actin dynamics, and result in an imbalance between members of the Rho GTPase family, and it appears to be the main explanation for the changes in synapse structure and stability reported in this work. Although the lack of Panx1 does not cause lethality, the overexpression of it is likely to contribute to neuronal pathologies, therefore the study is potentially an interest for a broad field of researchers. The analysis is complex and thorough, it involves the application of numerous techniques, the figures are carefully done, clarity of the ms is fine, and I believe that it is suitable manuscript for publication. Nevertheless, it remains a rather descriptive work, and I would highly appreciate more efforts towards mechanistic understanding. For example, what could be the basis of the differential effect on Rac1, cdc42 and RhoA? It is far from being trivial how this gap junction protein impacts on the expression level and activity of the Rho GTPases. An important experiment would be to investigate whether the observed defects and structural alterations are congenital or they develop with age and usage. A developmental analysis of at least some critical phenotypic components would make the paper much stronger.
Answer: We thank to the reviewer for the careful consideration of our manuscript and for the positive comments and suggestions that substantially improved the quality of our manuscript. We agree with the reviewer that the developmental analysis is an important component to address interesting mechanistic possibilities, but unfortunately, we cannot include this in the revised manuscript. instead, those experiments need to be addressed in future works.
We have not further addressed the mechanism of the differential effect on Rac1, Cdc42, and RhoA in our model by which these Rho GTPases regulate Panx1 channels. However, evidence suggest the involvement of RhoA in the activation of Panx1. Therefore, we include a new paragraph in the revised version of the manuscript discussing this point (page 28, lines 967-979).
Minor issues:
Although the use of English is fine in most parts of the paper, several sentences should be modified:
line 748: “operate compensatory mechanisms” was perhaps meant to be “compensatory mechanisms operate”.
Answer: Revised accordingly. Page 25, line 771.
line 800: “regulate tune” makes no sense
Answer: Revised accordingly. Page 26, line 823.
lines 819 and 831: “cannot discard” should be “cannot exclude”.
Answer: Revised accordingly. Page 26, line 842.
line 900: “the Arp2/3 complex involved in de novo nucleation”. The Arp2/3 complex, by definition, is not a de novo nucleation factor as it only works in the presence of an existing filament.
Answer: Suggestion accepted. This sentence has been modified. Page 28, line 927.
line 914: “in the actin dynamic process” makes no sense.
Answer: Revised accordingly. Page 28, line 943.
lines 916-918: “Activation of Rho GTPases and interaction with numerous ABPs and microtubule-associated proteins extensively interact and give feedback controlling their organization [103,104].” This sentence makes no sense.
Answer: Suggestion accepted. This sentence has been deleted.
line 955: “for proper structure and functioning of neural in hippocampal CA1 neurons”, this sentence makes no sense in this form.
Answer: Suggestion accepted. This sentence has been modified. Page 29, line 1003.
Reviewer 3 Report
The study by Flores-Munoz et al, reports the results of long-term pannexin 1 ablation on Rho GTPase activity and actin polymerization in hippocampal neurons. Overall, the manuscript is well written and tried to address complex actin polymerization processes in WT and KO pannexin-1 (PanX1) hippocampal neurons. I have the following comments for the authors:
1. The title of the manuscript refers to the effects on Rho GTPases and actin polymerization in the context of PanX1 in hippocampal neurons. The authors only showed their experiments on these key proteins in Figures 6 and 7. I suggest the authors should change the title to better reflect the outcomes of this work.
2. Though the authors cited reference 54 for measuring G-actin and F-actin in tissues and immunofluorescence, a panel that shows the application of G-actin marker probes needs to be included. The authors should clearly state which fluorescence marker for G-actin has been used.
3. While measuring actin (G and F-actin) levels in cells is possible using a kit from the cytoskeleton, accurate estimations from tissue lysates remain a challenge. The authors should use caution in the interpretation of their results as biased distribution of phalloidin to certain F-actin networks can result in artifacts. This limitation should be clearly mentioned in the manuscript. The authors should also clearly state in the manuscript that the accurate measurement of G- and F-actin levels in tissues require techniques such as two-photon fluorescence anisotropy imaging.
4. Explain the benefits and pitfalls of the PanX1 transgenic line in the discussion in terms of the actin cytoskeleton. Why is the constitutive global deletion of PanX1 is necessary?
5. The increased levels of Arp3 shown in Panx1-KO samples are quite intriguing. However, the differential expression levels of active RhoA shown in Panx1-KO tissues in Fig. 7 D-F are inconclusive. This should be addressed in the manuscript.
6. The order of Figure 7 legend labels F and G do not match the figure and should be corrected.
7. Numerous typos need to be corrected in the methods. Also, formatting throughout the manuscript should be checked for consistency.
Author Response
Reviewer 3
Comments and Suggestions for Authors
The study by Flores-Munoz et al, reports the results of long-term pannexin 1 ablation on Rho GTPase activity and actin polymerization in hippocampal neurons. Overall, the manuscript is well written and tried to address complex actin polymerization processes in WT and KO pannexin-1 (PanX1) hippocampal neurons. I have the following comments for the authors:
Answer: We really appreciate the reviewer’s comments and suggestions.
The title of the manuscript refers to the effects on Rho GTPases and actin polymerization in the context of PanX1 in hippocampal neurons. The authors only showed their experiments on these key proteins in Figures 6 and 7. I suggest the authors should change the title to better reflect the outcomes of this work.
Answer: We have changed the title of our manuscript as follows: “The Long-term Pannexin 1 ablation produces structural and functional modifications in hippocampal neurons”.
Though the authors cited reference 54 for measuring G-actin and F-actin in tissues and immunofluorescence, a panel that shows the application of G-actin marker probes needs to be included. The authors should clearly state which fluorescence marker for G-actin has been used.
Answer: Our intention with Figure 7A was to analyze the F-actin content in our experimental model, thus, we did´nt any G-actin marker in these experiments. However, the F/G actin commercial kit de Cytoskeleton includes G-actin standards that we have used to evaluate the sensibility of the test. In the future we will explore the impact of the Panx1 ablation in the actin cytoskeleton dynamics using fluorescent G-actin monomers.
While measuring actin (G and F-actin) levels in cells is possible using a kit from the cytoskeleton, accurate estimations from tissue lysates remain a challenge. The authors should use caution in the interpretation of their results as biased distribution of phalloidin to certain F-actin networks can result in artifacts. This limitation should be clearly mentioned in the manuscript. The authors should also clearly state in the manuscript that the accurate measurement of G- and F-actin levels in tissues require techniques such as two-photon fluorescence anisotropy imaging.
Answer: We thank the reviewer for pointing this out, which requires further research. However, we need to clarify that our interest was to explore the neuronal morphological changes that depends on the actin cytoskeleton dynamics, not to study the dynamics of actin polymerization per se.
Unfortunately, we do not have access to the equipment necessary for more accurate techniques such as two-photon for anisotropy analysis of the actin dynamics. However, we will certainly consider it for future studies. As requested, we’ve reduced the strength of the claims made by interpreting the results. We include a paragraph in the discussion section indicating the limitations of the methodology used (Page 27, lines 918-922).
Explain the benefits and pitfalls of the PanX1 transgenic line in the discussion in terms of the actin cytoskeleton. Why is the constitutive global deletion of PanX1 is necessary?
Answer: We appreciate the reviewer’s discussion in this regard. Indeed, we are aware that constitutive gene deletion can have some problems that may affect the correct interpretation of the results. In particular, constitutive deletion of the Panx genes had produced some contradictory results between different groups, which can be a consequence of incomplete penetrance of the genetic deletion and/or a compensatory over-expression of other related proteins, like Panx3, Cx26 or Cx30 (Abitbol, et al., 2016; Abitbol, et al., 2019). However, our previous findings using this Panx1 KO mouse, which is characterized by alterations in the long-term synaptic plasticity and cognitive function (Ardiles et al., 2014; Gajardo et al., 2018) have been reproduced using other animal models (Prochnow et al. 2012) and pharmacological blockage of Panx channels (Ardiles et al., 2014). In the present study, we wanted to explore if those effects could be due to modifications in the establishment of the neuronal circuits, specifically in the hippocampus. Therefore, we added an explanation for using our global Panx1-KO mouse model in the current version of our manuscript (Page 24, lines 759-768).
- Abitbol, J.M.; Kelly, J.J.; Barr, K.; Schormans, A.L.; Laird, D.W.; Allman, B.L. Differential effects of pannexins on noise-induced hearing loss. Biochem J 2016, 473, 4665-4680.
- Abitbol, J.M.; O'Donnell, B.L.; Wakefield, C.B.; Jewlal, E.; Kelly, J.J.; Barr, K.; Willmore, K.E.; Allman, B.L.; Penuela, S. Double deletion of Panx1 and Panx3 affects skin and bone but not hearing. J Mol Med (Berl) 2019, 97, 723-736.
- Ardiles, A.O.; Flores-Munoz, C.; Toro-Ayala, G.; Cardenas, A.M.; Palacios, A.G.; Munoz, P.; Fuenzalida, M.; Saez, J.C.; Martinez, A.D. Pannexin 1 regulates bidirectional hippocampal synaptic plasticity in adult mice. Front Cell Neurosci 2014, 8, 326.
- Gajardo, I.; Salazar, C.S.; Lopez-Espindola, D.; Estay, C.; Flores-Munoz, C.; Elgueta, C.; Gonzalez-Jamett, A.M.; Martinez, A.D.; Munoz, P.; Ardiles, A.O. Lack of Pannexin 1 Alters Synaptic GluN2 Subunit Composition and Spatial Reversal Learning in Mice. Front Mol Neurosci 2018, 11, 114.
- Prochnow, N.; Abdulazim, A.; Kurtenbach, S.; Wildforster, V.; Dvoriantchikova, G.; Hanske, J.; Petrasch-Parwez, E.; Shestopalov, V.I.; Dermietzel, R.; Manahan-Vaughan, D.; et al. Pannexin1 stabilizes synaptic plasticity and is needed for learning. PloS one 2012, 7, e51767.
The increased levels of Arp3 shown in Panx1-KO samples are quite intriguing. However, the differential expression levels of active RhoA shown in Panx1-KO tissues in Fig. 7 D-F are inconclusive. This should be addressed in the manuscript.
Answer: We now include a paragraph in the discussion section arguing about the potential relationship between Arp 3, Panx1, and RhoA (Page 28, lines 926-932).
The order of Figure 7 legend labels F and G do not match the figure and should be corrected.
Answer: We apologize if our original Figure 7 caption did not explain the F and G labels. We have modified the legend in the revised version of the manuscript (new figure 8).
Numerous typos need to be corrected in the methods. Also, formatting throughout the manuscript should be checked for consistency.
Answer: Thanks for your comments. We went through the entire manuscript to update the typo and text consistency.
Reviewer 4 Report
Pannexins, channel-forming glycoproteins, play an important role in intercellular communication. Pannexin channels dysfunction have been reported in several diseases including Alzheimer’s disease. Understanding the molecular mechanisms by which these channels influence neurodegeneration could lead to new therapeutic strategies. In this manuscript, Flores-Muñoz et al. report that the absence of pannexin 1 (Panx1) in hippocampal neurons of mice enhances neuronal excitability due the increase in the complexity of the dendritic network and the number of synaptic contacts, as a consequence of actin reorganization by the upregulation of Rac1 and downregulation of RhoA GTPase activity. The authors suggest a novel interaction between Panx1 channels, actin, and Rho GTPases for controlling neuronal morphology and functionality. My comments and suggestions below:
Major concerns:
The analysis shown in Figure 7 must be performed in isolated neurons. The F-actin intensity in Panx1-KO primary hippocampal neurons shown in Figure 6A/S3 seem to be similar or lower than in WT cells. Quantification is required. Show EGFP instead of merge in 6A (as in S3), replace S3 with alternative representative images, and display a heatmap for rhodamine-phalloidin intensity. In addition, direct evidence of Rac1 and RhoA activity in primary WT vs. Panx1-KO neurons must be shown (e.g., specific antibodies for the active form and FRET-based biosensors)
The morphological analysis of dendritic spines is questionable (e.g., in Figure 4H/K, the shape of top f spine considerably differs to bottom f spine and is comparable to t spine in Panx1-KO neurons). Further analyses (i.e., 3D rendering and clusterization) are required to be precise. To quantify, compare two categories: mature (m+c counts) vs. immature spines (f+t+s). Same applies to Figure 6H. Double-check bar size in panel 4H and 6E, spine lengths do not correspond to values shown in 4J and 6F, respectively.
The average PSD length quantified in Figure 5C differs to what is shown in representative TEM images (5A, S1A). Clarify the method for quantification; sum of multiple PSD per bouton?. The quality of TEM pictures is low, synaptic vesicles profiles are hard to visualize, include images at higher magnification/resolution and quantify SVs density per area unit.
Minor issues:
Material and Methods
Indicate the concentration of isoflurane and approximate time to anesthetize the animals
Review the EM protocol, embedding in LR white and Epon?
Double-check the catalog/product # of anti-Syntaxin, -Debrin, -WAVE 1, and -actin antibodies
Include antibodies used in the immunostaining section
Line 85; excitatory LTP and LTD => excitatory long-term potentiation (LTP) and depression (LTD)
Line 211; drawing device => tracing device
Line 229; length: width ratio => length/width ratio
Line 238; indicate the approximate volume of the tissue blocks
Line 261; m.o => month old (m.o.)
Lines 327, 399; exposure settings => image acquisition settings
Line 379; After that…, time?, …was aspirated from each well and added maintenance media... => …was replaced by growth media
Line 387; Define DIV abbreviation
Line 406; Normality distribution => Normality
Results
Double-check the levels of significance (*, **, ***) in graphs (Figures 1E, 1F, 2B, 2F, 5H, 5J, and 7G) vs. text/Figure legends.
Line 423; action potential => action potential (AP)
Line 427; significant differences are shown in Figure 1E
Include bar size in panel 4A
Figure 5 and S1; slices => ultrathin sections. Include the total number of synaptic boutons analyzed
Lines 691, 702, and 709; phalloidin => rhodamine-phalloidin
Typo & text errors
Line 300; anti-N WASP => anti-WASP
Line 377; planting => plating, SBF => FBS
Line 383 …,and maintained at 37°C => at 37°C
Line 393; …permeabilized with and permeabilized with => and permeabilized with
Line 421; RMP => Vm
Line 544; *p=0.0012 => **p=0.0012
Line 561; Figure 4D => Figures 4D, 4E
Line 670; EFGP => EGFP
Author Response
Reviewer 4
Comments and Suggestions for Authors
Pannexins, channel-forming glycoproteins, play an important role in intercellular communication. Pannexin channels dysfunction have been reported in several diseases including Alzheimer’s disease. Understanding the molecular mechanisms by which these channels influence neurodegeneration could lead to new therapeutic strategies. In this manuscript, Flores-Muñoz et al. report that the absence of pannexin 1 (Panx1) in hippocampal neurons of mice enhances neuronal excitability due the increase in the complexity of the dendritic network and the number of synaptic contacts, as a consequence of actin reorganization by the upregulation of Rac1 and downregulation of RhoA GTPase activity. The authors suggest a novel interaction between Panx1 channels, actin, and Rho GTPases for controlling neuronal morphology and functionality. My comments and suggestions below:
Answer: We thank the reviewer for their comments and suggestions that substantially will improve the quality of our manuscript.
Major concerns:
The analysis shown in Figure 7 must be performed in isolated neurons. The F-actin intensity in Panx1-KO primary hippocampal neurons shown in Figure 6A/S3 seem to be similar or lower than in WT cells. Quantification is required. Show EGFP instead of merge in 6A (as in S3), replace S3 with alternative representative images, and display a heatmap for rhodamine-phalloidin intensity.
Answer: We appreciate the reviewer’s suggestions. We separate the former Figure 6 in two figures (new figures 6 and figure 7). Figure 6 display a new panel showing the split channels for the rhodamine-phalloidin and EGFP fluorescence in the hippocampal neuron cultures. Figure 7 shows images and quantification of the rhodamine-phalloidin fluorescence intensity in dendritic segments. We also include a new supplementary figure S3 showing rhodamine-phalloidin fluorescence in hippocampal tissues.
In addition, direct evidence of Rac1 and RhoA activity in primary WT vs. Panx1-KO neurons must be shown (e.g., specific antibodies for the active form and FRET-based biosensors)
Answer: We agree with the reviewer that further experiments in isolated cultured neurons are needed on this point. However, these approaches could take a long time. Furthermore, currently, we cannot carry out experiments using FRET sensors, but certainly, we will consider it for future studies.
The morphological analysis of dendritic spines is questionable (e.g., in Figure 4H/K, the shape of top f spine considerably differs to bottom f spine and is comparable to t spine in Panx1-KO neurons).
Answer: Thank you very much for your valuable suggestion. This was an oversight. We apologize for our error. We corrected the figures 4H-K in the revised version of the manuscript.
Further analyses (i.e., 3D rendering and clusterization) are required to be precise. To quantify, compare two categories: mature (m+c counts) vs. immature spines (f+t+s). Same applies to Figure 6H. Double-check bar size in panel 4H and 6E, spine lengths do not correspond to values shown in 4J and 6F, respectively.
Answer: We apologize if our original Figure 4 did not properly show the length of the dendritic spines and the respective calibration bar. As visualized in our conditions, we agree with the reviewer that the classification of dendritic spines does not represent the most precise method to identify different shapes. However, we classified spines manually according to the parameters described in the literature for Golgi-stained neurons, including the size of the spine head, spine neck, spine length, and neck/length ratio (Richer et al., 2014). Nonetheless, despite the limitations of the process, they are not likely to substantially limit the quality of evidence. In the present study, we use fluorescent confocal microscopy due to the possibility of analyzing a higher number of spines (Nagerl et al., 2008; Bączyńska et al., 2021). But indeed, super-resolution fluorescent microscopy, two-photon microscopy, and EM should providemore detailed morphometric analysis at the nanoscale level. As suggested by the reviewer, now we grouped the spines morphology into immature vs. mature spines and included a new graph with the respective quantification in the revised version of the manuscript (Figure 4F and 7G).
-Risher, W.C.; Ustunkaya, T.; Singh Alvarado, J.; Eroglu, C. Rapid Golgi analysis method for efficient and unbiased classification of dendritic spines. PloS one 2014, 9, e107591
- Nagerl, U.V.; Willig, K.I.; Hein, B.; Hell, S.W.; Bonhoeffer, T. Live-cell imaging of dendritic spines by STED microscopy. Proc Natl Acad Sci U S A 2008, 105, 18982-18987, doi:10.1073/pnas.0810028105.
- Bączyńska, E.; Pels, K.K.; Basu, S.; Włodarczyk, J.; Ruszczycki, B. Quantification of Dendritic Spines Remodeling under Physiological Stimuli and in Pathological Conditions. Int. J. Mol. Sci. 2021, 22, 4053.
The average PSD length quantified in Figure 5C differs to what is shown in representative TEM images (5A, S1A). Clarify the method for quantification; sum of multiple PSD per bouton? The quality of TEM pictures is low, synaptic vesicles profiles are hard to visualize, include images at higher magnification/resolution and quantify SVs density per area unit.
Answer: We apologize for mistakenly showing the calibration bar and the spine length graph in figure 5. We replaced the representative TEM images of asymmetric synapses to show a little clearer synaptic structure. On the other hand, we detected some inconsistencies in the pixel to micrometers conversion by closer looking at the results obtained in the spine length. Therefore, we reanalyzed the data and the new measurements with the respective scale bar appear now in the new version of figure 5D.
Minor issues:
Material and Methods
Indicate the concentration of isoflurane and approximate time to anesthetize the animals Review the EM protocol, embedding in LR white and Epon?
Answer: Animals were deeply anesthetized with 5% isoflurane for 5 min and then decapitated when fully sedated, as measured by the lack of active paw reflex. We now included this description in the section 2.3 of materials and methods (page 3, lines 122-123).
Regarding the EM protocol, we’ve corrected the description of the procedure in the 2.11 section of material and methods (page 5, lines 236-240) in the revised version of the manuscript.
Double-check the catalog/product # of anti-Syntaxin, -Debrin, -WAVE 1, and -actin antibodies.
Answer: Suggestion accepted. The catalog/product has been corrected in the section Material and methods in the revised version of the manuscript.
Include antibodies used in the immunostaining section.
Answer: We did not make any immunostaining procedures. Cultured neurons were stained with rhodamine-phalloidin to visualize actin fibers cytoskeleton in transfected cells with EGFP constructs (empty vector and Panx1). We include an extra section describing this protocol. Page 8, lines 389-405.
Line 85; excitatory LTP and LTD => excitatory long-term potentiation (LTP) and depression (LTD).
Line 211; drawing device => tracing device
Line 229; length: width ratio => length/width ratio
Line 238; indicate the approximate volume of the tissue blocks
Line 261; m.o => month old (m.o.)
Lines 327, 399; exposure settings => image acquisition settings
Line 379; After that…, time?, …was aspirated from each well and added maintenance media... => …was replaced by growth media. Line 387; Define DIV abbreviation.
Line 406; Normality distribution => Normality.
Answer: Revised accordingly, and we’ve incorporated all suggestions made by the reviewer.
Results
Double-check the levels of significance (*, **, ***) in graphs (Figures 1E, 1F, 2B, 2F, 5H, 5J, and 7G) vs. text/Figure legends.
Line 423; action potential => action potential (AP)
Line 427; significant differences are shown in Figure 1E
Include bar size in panel 4A
Figure 5 and S1; slices => ultrathin sections. Include the total number of synaptic boutons analyzed.
Lines 691, 702, and 709; phalloidin => rhodamine-phalloidin.
Answer: Thank you for your comments. We have revised the significance levels in all figures and changed the legend text. In addition, we've incorporated all suggestions throughout the revised version of the manuscript.
Typo & text errors
Line 300; anti-N WASP =>anti-WASP
Line 377; planting => plating, SBF => FBS
Line 383 …,and maintained at 37°C => at 37°C
Line 393; …permeabilized with and permeabilized with => and permeabilized with
Line 421; RMP => Vm
Line 544; *p=0.0012 => **p=0.0012
Line 561; Figure 4D => Figures 4D, 4E
Line 670; EFGP => EGFP
Answer: We’ve corrected the typo and text errors in the revised version of the manuscript.
Round 2
Reviewer 2 Report
Although further experimental work is not included, the textual additions at numerous places highly increased the clarity of the manuscript. Most of the minor grammatical errors are also corrected, therefore I suggest to accept the paper in this form.
Reviewer 4 Report
The authors have reasonably addressed most of my comments.
Pending correction; scale bars size in Figures 4A, 4H, and K